# Automated high-throughput genome editing platform with an AI learning in situ prediction model

Siwei Li [1,2,5], Jingjing An[1,2,5], Yaqiu Li[1,2,5], Xiagu Zhu[1,2,3], Dongdong Zhao [1,2], Lixian Wang[1,2], Yonghui Sun[1,2,4], Yuanzhao Yang[1,2,3], Changhao Bi [1,2] ✉, Xueli Zhang [1,2] ✉ & Meng Wang [1,2,4] ✉

A great number of cell disease models with pathogenic SNVs are needed for the development of genome editing based therapeutics or broadly basic scientific research. However, the generation of traditional cell disease models is heavily dependent on large-scale manual operations, which is not only time-consuming, but also costly and error-prone. In this study, we devise an automated high-throughput platform, through which thousands of samples are automatically edited within a week, providing edited cells with high efficiency. Based on the large in situ genome editing data obtained by the automatic high-throughput platform, we develop a Chromatin Accessibility Enabled Learning Model (CAELM) to predict the performance of cytosine base editors (CBEs), both chromatin accessibility and the context-sequence are utilized to build the model, which accurately predicts the result of in situ base editing. This work is expected to accelerate the development of BE-based genetic therapies.

Single nucleotide variations (SNVs)[1] in the human genome may generate changes of transcription levels, protein sequences or many other properties of the original DNA, and cause genetic diseases. According to the ClinVar database of NCBI[2], more than 37,000 known diseases are associated with pathogenic SNVs. The diseases caused by single nucleotide mutations include rare diseases, such as sickle cell disease, thalassemia, and Leber congenital amaurosis. A study by the European Organization of Rare Diseases has shown that more than 450 million people worldwide suffer from rare diseases, 95% of which remain without effective treatments to date[3]. Even in some diseases with therapeutic solutions, patients usually require lifelong medicine while having low-quality life, and a shorter life expectancy[4].

The CRISPR/Cas9 technology is considered an ideal system to investigate and treat genetic diseases[4], infectious diseases, cancers, and immunological diseases[5,6]. As a newer-generation CRISPR technology, base editors (BEs), enable direct, irreversible correction of base mutations, which have a promising future for curing genetic diseases caused by SNVs. Compared with standard genome editing, BEs can effectively repair base mutations without inducing double-stranded DNA breaks (DSBs), which reduce the occurrence of insertions or deletions (indels) at target sites. Three classes of base editors have been reported, including cytosine base editors (CBEs)[7,8] that induce C•G to T•A conversion, adenine base editors (ABEs)[1] that induce A•T to G•C conversion, and glycosylase base editors (GBEs)[9,10] with C•G to G•C conversion.

These BEs provide almost ideal solutions for treating more than half of known pathogenic SNVs. However, before BE-based genetic therapies can be implemented, it is necessary to construct mammalian cell disease models for developing and optimizing BEs and enabling applications in gene therapy[11,12]. Due to the great number of genetic diseases and known SNVs, it is necessary to develop a method that enables the construction of a large number of cell models carrying different pathogenic SNVs, so that extensive research on gene therapies can be performed to find curing solutions. According to ClinVar,

[1]Tianjin Institute of Industrial Biotechnology, Chinese Academy of Sciences, Tianjin, China. [2]Key Laboratory of Systems Microbial Biotechnology, Chinese Academy of Sciences, Tianjin, China. [3]College of Biotechnology, Tianjin University of Science and Technology, Tianjin, China. [4]School of Life Sciences, Division of Life Sciences and Medicine, University of Science and Technology of China, Hefei, China. [5]These authors contributed equally: Siwei Li, Jingjing An, Yaqiu Li. ✉e-mail: bi_ch@tib.cas.cn; zhang_xl@tib.cas.cn; wangmeng@tib.cas.cn

approximately 50% of total human pathogenic SNVs are C•G to T•A conversion, which can be corrected by ABEs[13]. However, it is currently hard to obtain large numbers of cell models carrying these SNVs with reasonable labor and funding investment.

Mammalian cell lines are usually used for developing and optimizing the BEs, and research for prediction of base editing performance, such as efficiency and specificity, requires a large amount of editing data. To solve this problem, methods based on a target-locus integration library were developed, such as the Be-Hive, which provided the data for AI to learn and predict the editing performance[14]. However, currently such data was obtained from integrated editing loci which lacked in situ information[15]. Previous research has shown a strong correlation between the performance of nuclease and the chromatin accessibility properties[14,16]. Kristopher et al. demonstrated that gene editing was more efficient in euchromatin than in heterochromatin[16]. Large-scale genetic screens in human cell lines indicated that highly active sgRNAs for Cas9 and dCas9 were found in regions of low nucleosome occupancy, and the nucleosomes directly impeded Cas9 binding and cleavage in vitro[17]. We previously found that pioneer factor, such as Vp64, improved CRISPR-based C-to-G and C-to-T base editing by changing local chromatin environment[18]. However, current studies on deep learning employed the editing data from lentiviral integrated target sequences, while the real chromosomal environment of the target sequence was ignored[13,14]. One of the reasons is that it is difficult to obtain a large set of editing data from endogenous target sites. For large scale samples, manual operations are not only time-consuming, but also error prone, less consistency and expensive[19,20]. An automatic platform would make it possible to get large-scale editing dataset of endogenous targets. And with the large-scale in situ editing data and sequence information, combined with local chromatin accessibility, a machine learning model with in situ data might be able to better predict the actual base editing efficiency.

In this study, we devise an automated platform that performs the whole genome editing process from guide RNA (gRNA) design to the analysis of the editing results, which comprehensively characterizes the relationships of the in situ base editing outcomes with the sequence and chromatin environment for BEs.

## Results

### Development of an automated platform for genome editing in mammalian cells

To increase the standardization and scalability of gene editing in mammalian cells, we designed an automated platform for high-throughput gene editing. The automated platform was mainly constituted of four modules, including (1) computer-aided design of endogenous target gRNAs, (2) the construction of gRNA expression plasmids, (3) base editing in mammalian cell, and (4) machine learning for CBEs performance model building, which is shown in Fig. 1. The equipment employed in the automated platform included the acoustic liquid handler, the plate sealer, the centrifuge, the automated thermocycler, the automated colony picker, the incubator shaker, and the liquid handler.

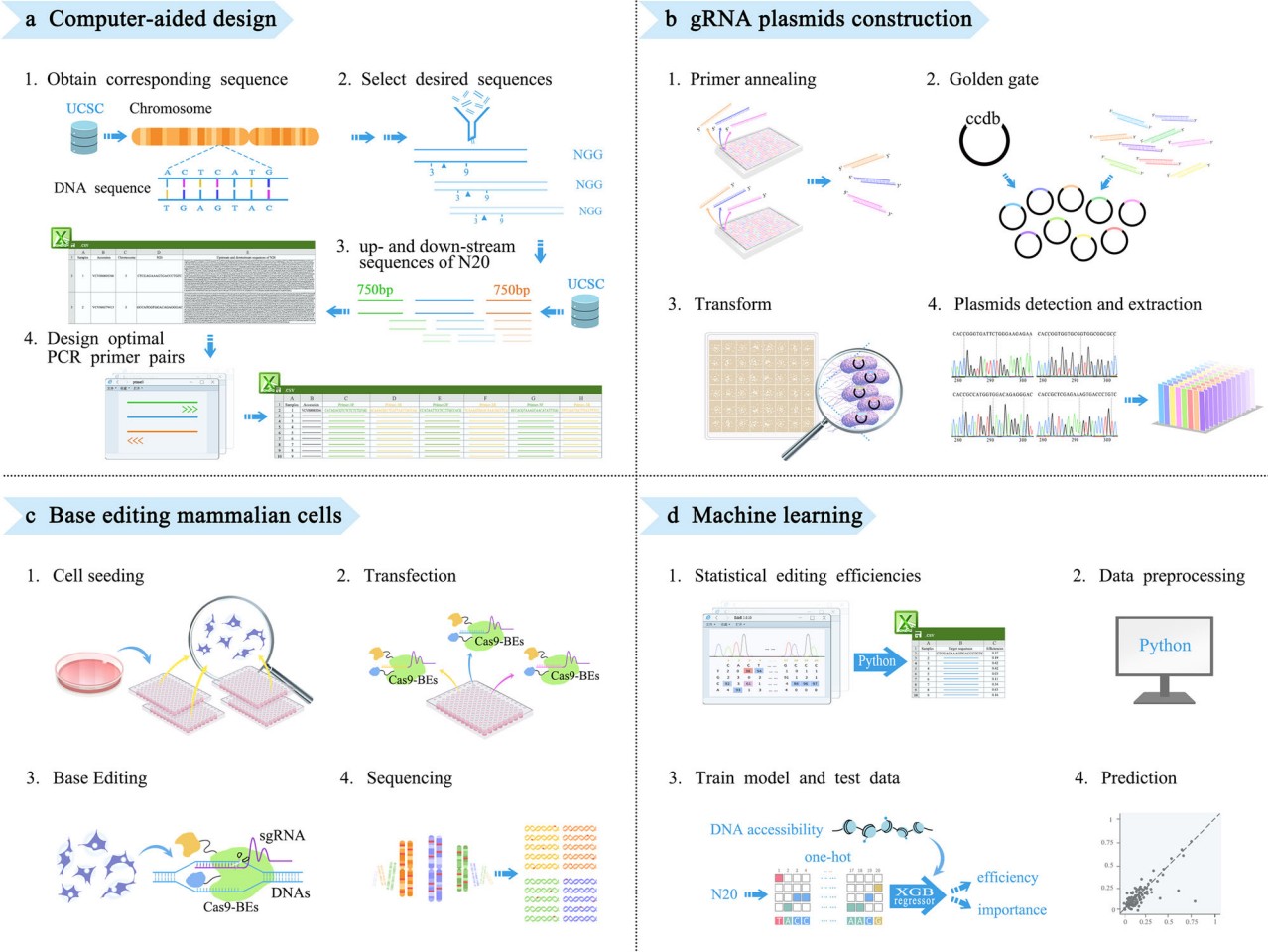

**Fig. 1 | Overview of automated high-throughput platform of gene editing in mammalian cells. a** Machine design module, gRNA plasmids design and batched primers acquisition. **b** gRNA plasmids construction module. **c** Cellular base editing module. **d** Machine learning of the in situ editing results.

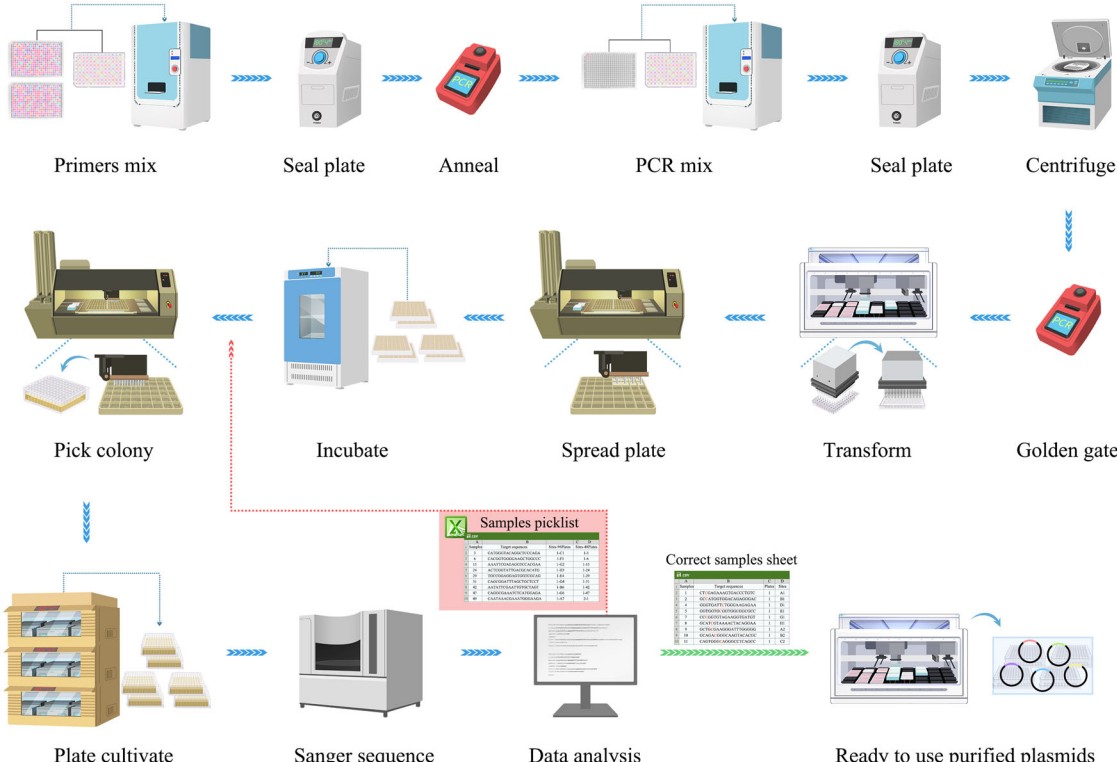

**Fig. 2 | Schematic Overview of the automated gRNA plasmids construction workflow.** The detailed procedures and equipment are listed. The red arrow means the incorrect information of assembled samples, and more colonies need to be picked for verification. The green arrow means correct information of assembled samples, and plasmid extraction can be made downstream.

The first module (Fig. 1a) was responsible for designing gRNAs in order to introduce the human pathogenic SNVs into wild-type cells. Tabular format data from ClinVar (NCBI)[2] was employed for querying C-to-T or G-to-A SNVs and associated information, including the locations and gene names. DNA segments encompassing the SNV loci were retrieved from NCBI's nucleotide database using the Python package seqseek. A 20-nt subsequence of the segment was selected to serve as a gRNA spacer of the target sequence with the following conditions. 1) It contained the SNV within the editing window spanning positions 3 to 9, and without another C, 2) and followed by a 3-nt NGG PAM sequence. Based on the ClinVar database, 1210 genes were selected as the target sites by bioinformatic analysis (the sequences of the protospacer for BE4max are listed in Supplementary Data 1, and the corresponding primer pairs for gRNAs construction are shown in Supplementary Data 2). Three pairs of primers for the analysis of editing results (Supplementary Data 3) were processed in batches, using the DNA sequence encompassing the target region from 750 bp upstream to 750 bp downstream of each target sites. Primer design was based on the GRCh37 chromosome sequence, employing a public primer3 server (https://bioinfo.ut.ee/primer3/) through batch queries by a Python script.

For the second module (Fig. 1b), schematic overview for the automated gRNA plasmids construction workflow is illustrated in Fig. 2. The acoustic liquid handler (Echo) was used for manipulating the DNA assembly reactions. DH5α competent cells were mixed with Golden Gate reaction products by a Beckman i7 liquid handler. Transformant plating was then prepared using an ClonePix™ system. The constructed plasmids were verified by DNA sequencing, and the correct plasmids were extracted by Beckman i7 liquid handler. For the analysis of data editing results, a Python script was used to read the sanger sequencing files, compare N20, and create two reference csv files. The false assembly csv file included a picklist for picking new colonies from 48-well bacterial colony plate to 96-well deep plate for

another round of verification by ClonePix™ system (Supplementary Data 4). The correct assembly csv file contained N20 sequencing and their location in 96-well deep plate, for plasmid extraction with Beckman i7 liquid handler (Supplementary Data 5). A total of 1210 gRNA plasmids were constructed and analysed by the high-throughput automatic system within 4 days with a successful rate of 99%, achieving a throughput of 384 gRNA constructs per day. The Beckman i7 lliquid handler was then used for plasmid extraction with a throughput of 576 plasmids per day.

The third module was the base editing in mammalian cells (Fig. 1c). By optimizing the experimental conditions, we developed an editing process including cell seeding and transfection, medium exchange of the cell culture, and sample collecting by the Beckman i7 liquid handler. The subsequent steps, including cell lysis and PCR amplification of the target region, were performed by an automated thermocycler. The details of the workflow of the automated gene editing are shown in Fig. 3. The liquid handler Beckman i7 was primarily used in this procedure. 1210 sets of gRNAs and BE4max plasmids were co-transfected into HEK293T cells using the automatic high-throughput system within 6 h, and the subsequent medium exchange was completed within 2 h. After 5 days of cultivation, the edited cells were harvested for PCR analysis, which were finished within 8 h, followed by the sanger sequencing. EditR software was used for quantifying the base editing efficiency from the sanger sequencing data, which is a free online tool broadly employed in the field[21]. Using the sanger sequencing files and corresponding gRNA sequences respectively, editing results can be acquired in batches by another Python script. For the analysis of editing efficiencies in batches, using Python script, we generated three csv files, one was a picklist for preparing a new round of PCR to analyse the false samples from 96-well lysis sample plates to 96-well PCR plates by Beckman i7 liquid handler (Supplementary Data 6). The second csv file contained the sequence and position

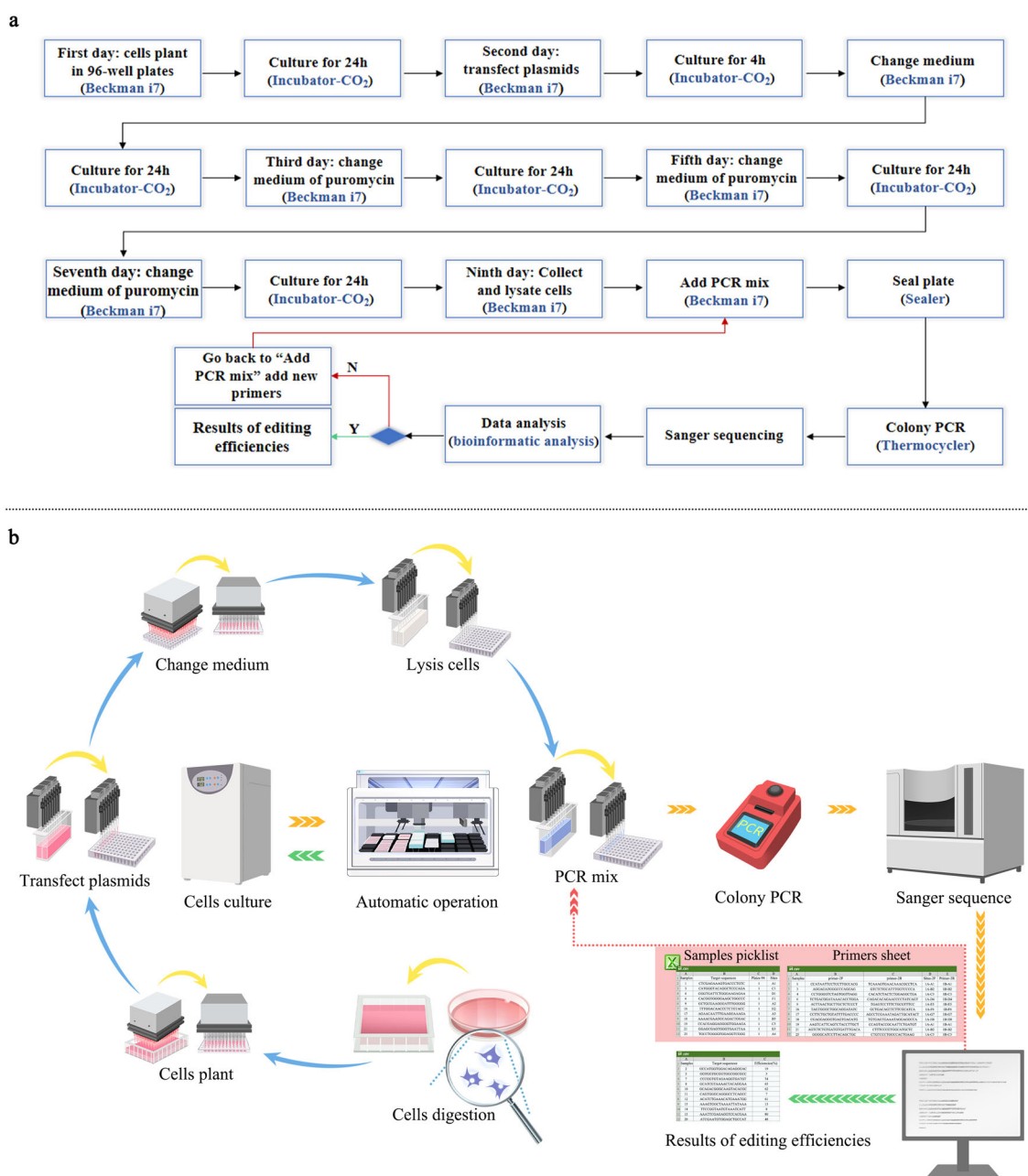

**Fig. 3 | Overview of the workflow of the automated gene editing process. a** A flowchart showing different instruments to orchestrate the gene editing process. The whole process starts from cultivating cells in 96-well plates, and ends at validation of editing efficiencies. The red arrow means the wrong PCR samples' information on sanger sequence, and new primers need to be used for PCR. The green arrow means correct PCR samples' information on sanger sequence, and editing efficiencies can be made downstream for AI learning. The steps of experiments are listed in black characters, and the corresponding instrument are listed in blue. The detailed protocols are available in Supplementary Fig. 1. **b** Workflow of the automated gene editing process in mammalian cells with details. The red arrow means the wrong analysis samples' information, and sample's information and new pair primers need to be picked for PCR. The green arrow means correct analysis samples' information and editing efficiencies, and the results of editing efficiencies can be made downstream for AI learning.

information of the second primer pair for the analysis of false samples. These primer pairs were synthesized in 96-well primer plate for a new round of PCR (Supplementary Data 7). The third csv file contained the correct samples and the editing efficiencies results for next step of AI learning (Supplementary Data 8). Figure 2 and Fig. 3 illustrate the platform components and the automatic workflows.

The fourth module was the machine learning (Fig. 1d), an AI model was developed by the analysis of the editing results, to predict the outcomes of base editing. The XGB Regressor module of the XGBoost classifier[22] was employed to predict the BE4max editing efficiency, based on information of the 20 bp protospacer sequence and the DNA accessibility value.

In summary, we developed an automated platform for high-throughput in situ gene editing of mammalian cells, allowing the parallel processing of a considerable number of samples, which simplified the repetitive and labour-intensive manual laboratory work and provided high-quality data for further machine learning. Detailed workflow of each module, the inputs, outputs, and human intervention steps in each module are shown in supplementary Fig. 1 and Fig. 2. By

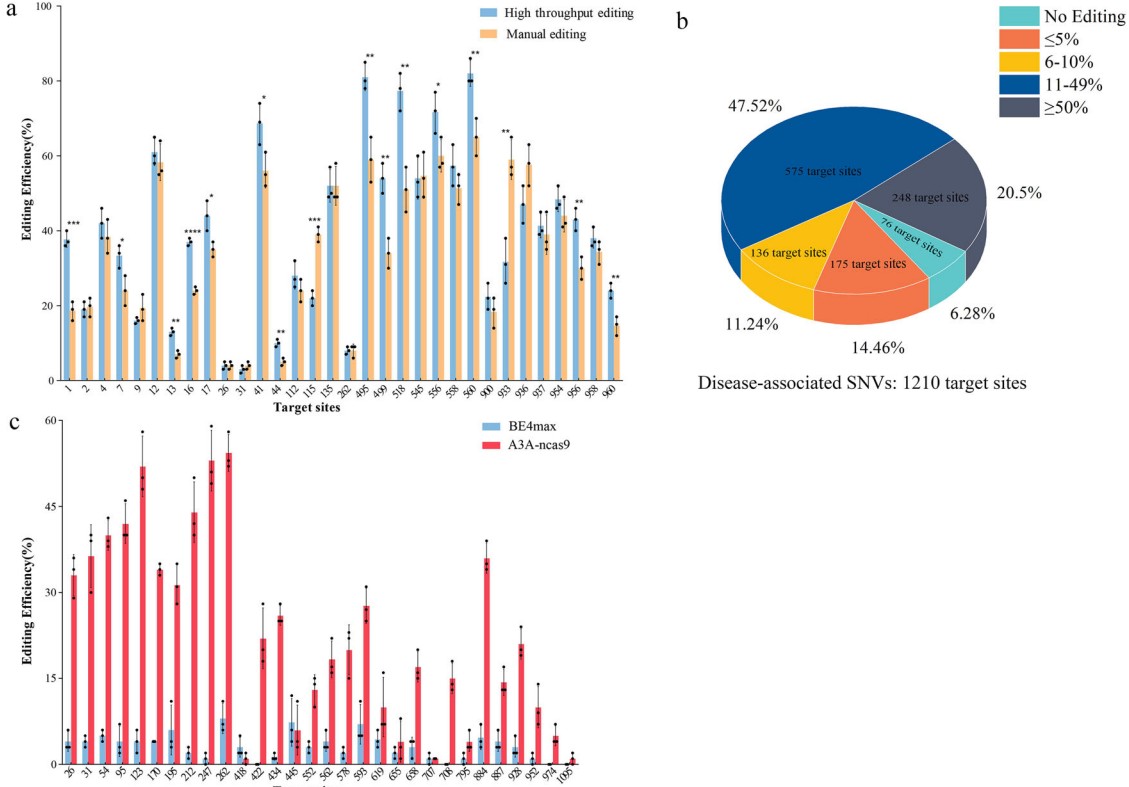

**Fig. 4 | Performance analysis of the automated high-throughput gene editing platform for mammalian cells. a** Comparison of the base editing efficiency of the automatic high-throughput system versus the manual operation at 32 genomic loci. Values and error bars reflect the mean ± SEM of three biological replicates ($n = 3$). The two-tailed Student's t-test for comparison of the two operations.*$p < 0.05$, $^{**}p < 0.01$, $^{***}p < 0.001$, $^{****}p < 0.0001$ compared to Manual editing. **b** Characterization of the high-throughput gene editing system at 1210 endogenous genomic loci. **c** Comparison of the editing efficiencies of BE4max and APOBEC3A-nCas9 at 30 genomic loci.

comparation, we found that high-throughput was superior to manual operation, both in terms of time and cost. With the increase of sample size, the advantage of automated high-throughput platform will be more significant. The detailed comparisons are supplied in supplementary Fig. 3.

## The editing efficiency of the automated high-throughput platform

To determine whether the editing efficiency data generated by the automated high-throughput system was similar to that of manual operation, we compared it with that of the manual operation at 32 different genomic loci. As shown in Fig. 4a, 16 target sites showed nearly equal editing efficiencies by the two operational processes, while the automated high-throughput system exhibited higher editing efficiencies at the other 14 sites by statistical analysis. The mean editing efficiency values of 32 targets, including the high-throughput system and manual manipulation, are listed in supplementary Table 1. These results suggested that the automated high-throughput system was able to perform base editing with the efficiencies comparable to those of the manual operations.

To demonstrate the capacity of the automated high-throughput system, a total of 1210 disease-associated SNVs were selected randomly as targets for editing with BE4max. Among the 1210 targets, 823 showed 10 to 50% C-to-T conversion efficiency, 248 targets had ≥ 50% editing efficiency, and 136 targets showed 5 to 10% editing efficiency. We were unable to obtain editing results from 76 gene targets due to unsuccessful PCR amplification of the target loci, and 175 gene targets had less than 5% efficiencies (Fig. 4b). The editing efficiencies of the 1210 targets are listed in the Supplementary Data 9. The editing results of BE4max were in accordance with the previous

reports, which showed greatly varied performance at different target sites[23].

By analyzing the sequence of the 175 loci with low efficiencies, we found that most of the targeted Cs are in the context of CG or GC nucleotides. Therefore, APOBEC3A-nCas9[24], which can mediate efficient C-to-T base editing in regions with high methylation levels, was introduced for further editing of 30 gene loci randomly chosen from the 175 loci. In this pool, 23 gene loci were edited with higher editing efficiencies by APOBEC3A-nCas9 with a mean editing frequency of 29.1%, while BE4max only achieved an average editing efficiency 3.4% (Fig. 4c). The 30 targets sequence and the mean values of efficiencies for BE4max and APOBEC3A-nCas9 are shown in supplementary Table 2. These results indicated that the unsuccessful editing of some targets might not be caused by the automatic high-throughput platform. Thus, the automatic high-throughput platform for human cell was successfully established, and thousands of endogenous target sites can be manipulated simultaneously and efficiently.

## Developing an AI learning model for predicting the base editing efficiency based on in situ data

In this work, with the automated high-throughput genome editing system, we obtained a large-scale data of in situ editing in mammalian cell. The datasets have high uniformity and repeatability, which could not be accessed easily by manual operation.

XGBoost Regressor[22], which is a popular machine learning model due to its advantages in many aspects including flexibility and regularization, was used to construct machine learning models for predicting the in situ base editing results of human cell lines. We previously utilized convolutional neural networks (CNN)[25] to predict GBE base editing efficiency, but we selected XGB Regressor in this

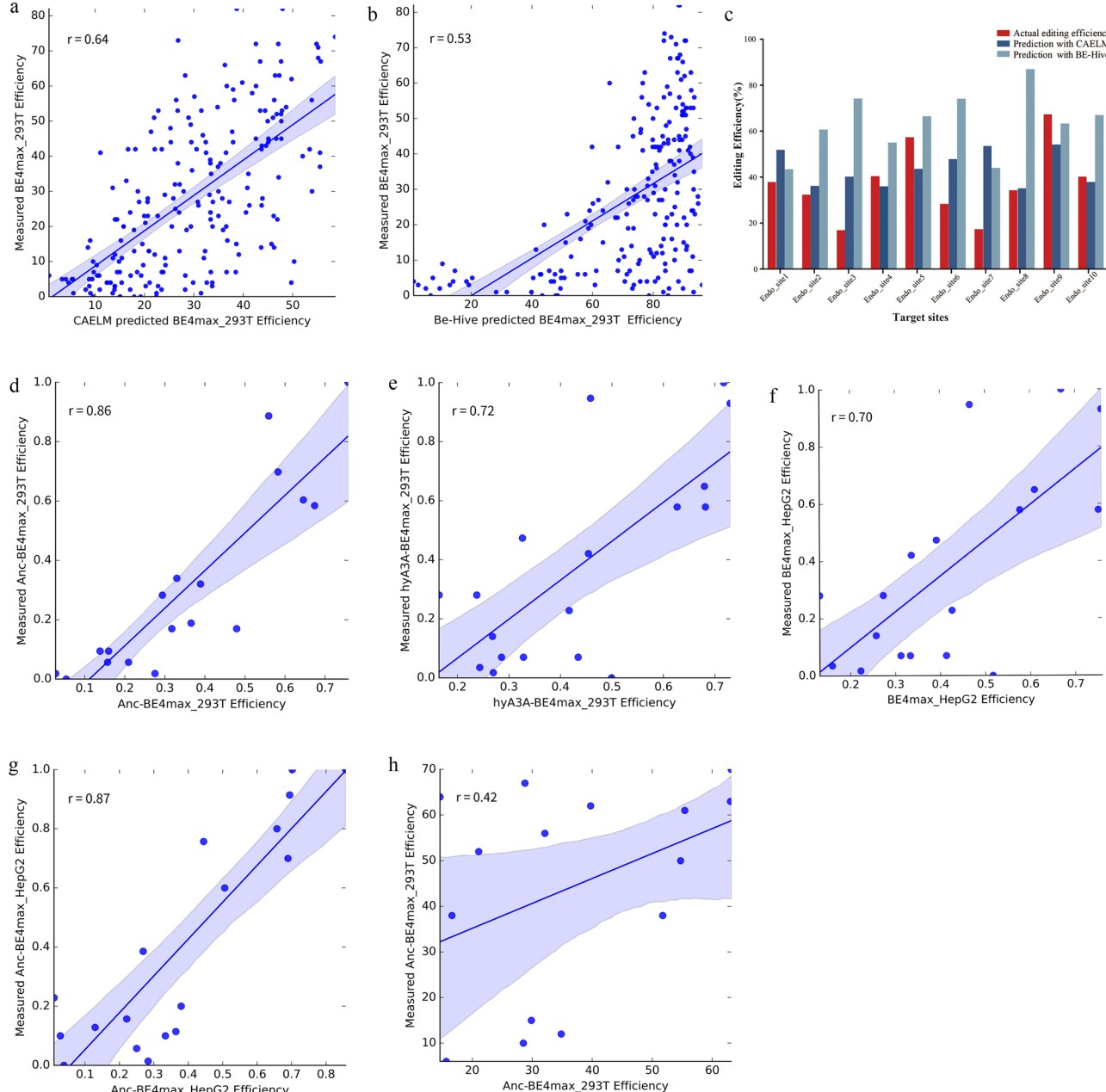

**Fig. 5 | The performance of the Chromatin Accessibility Enabled Learning Model (CAELM) for prediction base editing outcome.** Linear regression plots of fig. 5a, b, d–h. are presented with a 95% confidence-interval band around the regression line **a** Correlation between the editing efficiency predicted by XGB Regressor versus the actual values for BE4max.The Pearson's correlation coefficient (r) is shown. **b** Comparison of BE-Hive predicted (x-axis) and actual base editing efficiencies (y-axis). The Pearson's correlation coefficient(r) is shown. **c** The performance comparison of the CAELM and BE-Hive. **d** Correlation between the editing efficiency predicted by XGB Regressor versus the actual values for Anc-

BE4max in HEK293T cells. **e** Correlation between the editing efficiency predicted by XGB Regressor versus the actual values for hyA3A-BE4max in HEK293T cells. **f** Correlation between the editing efficiency predicted by XGB Regressor versus the actual values for BE4max in HepG2 cells. **g** Correlation between the editing efficiency predicted by XGB Regressor versus the actual values for Anc-BE4max in HepG2 cells. **h** Correlation between the editing efficiency predicted by XGB Regressor versus the actual values for hyA3A-BE4max in HepG2 cells. The Pearson's correlation coefficient (r) is shown.

study, because our dataset only consisted of editing results for 1134 target sequences, and the XGB Regressor commonly exhibits better performance than the deep neural networks when dealing with small datasets[22]. Based on the above 1134 valid editing results, with the context-sequence information and chromatin accessibility value of each endogenous target sites, we constructed machine learning models to predict the in situ base editing results of human cell lines. CAELM was designated as the Acronym for the Chromatin Accessibility Enabled Learning Model (CAELM) in this work. The model predicting the base editing efficiencies of BE4max in HEK293T was trained and

tested in an 80:20 ratio, meaning 80% targets information was used to train the model, and the remaining 20% was used for prediction testing. The accuracy of the model was measured by Pearson's correlation coefficient[26], and achieved a r value of 0.64 between the measured and predicted value (Fig. 5a). We also performed a 5 × 5 nested cross-validation[27], which is a gold standard for reliably performing hyper-parameters tuning and performance assessment. We used an inner loop to select optimal hyperparameters by grid search and an outer loop to evaluate the performance of the model (Supplementary Fig. 5a). Concretely, the data was divided into 5 outer folds, each fold

accounted for 20% and was left out as the testing fold. The remaining 80% samples were further split into 80% training and 20% validation set in the inner loop to obtain the optimized hyperparameters, which were used in the outer loop to retrain and evaluate the model, a Pearson's correlation coefficient, $r = 0.64$, was obtained as the average metric value across the five outer testing folds and suggested a good prediction accuracy (Supplementary Fig. 5b). In the machine learning process, the average values of DNaseI HS Density Signal over the protospacer sequences of the edited loci were retrieved from Encyclopedia of DNA Elements (ENCODE)[28] with UCSC accession numbers, and used as the chromatin accessibility parameter for input, based on the HEK293T GRCh37/hg19 human reference genome.

## Comparison of prediction accuracy between CAELM and BE-Hive

The BE4max editing efficiencies of the 209 testing loci were also predicted using the public prediction tool BE-Hive (http://www.crisprbehive.design)[15]. Since the tool only accepts the sequence information, DNA sequences containing 20 nt on the 5′ side and 10 nt on the 3′ side flanking the N20 sequence were fed into BE-Hive. The predicted outputs were then compared with the actual in situ base editing efficiencies we obtained, and the Pearson's coefficient r value was calculated. Compared with CAELM, a relatively lower correlation was observed between the BE-Hive predicted and the actual base editing efficiencies ($r = 0.53$) (Fig. 5b). Furthermore 10 endogenous target sites retrieved from previous report[14] of third party were obtained, and used for prediction analysis by CAELM and BE-Hive respectively. The actual editing efficiencies and predicted results are shown in Fig. 5c, and the sequence and editing efficiencies of endogenous target sites are listed in Supplementary Table 3. The results suggested that CAELM, trained with in situ data, outperformed the current prediction model by utilizing both sequence and chromatin information as input. Although currently we are not able to provide the web service, the prediction model was uploaded to GitHub (https://github.com/YQLiCAS/BE4max) for public researching usage.

## Expanding the application scope of CAELM for different base editors and cell lines

To further enhance the application value of CAELM, in addition to BE4max, two other typical CBEs, Anc-BE4max and hyA3A-BE4max, were added in the model learning. To consider the difference of chromatin accessibility parameter in different cell types, the C-to-T conversion of about 100 endogenous target sites were analysed in HEK293T and HepG2 cells respectively (Supplementary Fig. 4). The combinatorial experiments generated 5 corresponding data sets, including Anc-BE4max_HEK293T, hyA3A-BE4max_HEK293T, BE4max_HepG2, Anc-BE4max_HepG2 and hyA3A-BE4max_HepG2 (Supplementary Data 10).

To boost the fitted model on new data, we continued training via loading the predictive model of BE4max_293T using the parameter 'xgb_model = ' in the model fitting process. Five models including Anc-BE4max and hyA3A-BE4max in HEK293T, BE4max, Anc-BE4max and hyA3A-BE4max in HepG2 were built by continuing learning on the BE4max_293T model. The new data was continually trained on the BE4max_293T model and tested at an 85/15 ratio. Data of the target sequences shared with the test set was excluded from the data of BE4max in HEK293T to train the base model in the first place. Good correlations were achieved between the experimental and predicted efficiencies of Anc-BE4max (Pearson's $r = 0.86$), hyA3A-BE4max in HEK293T (Pearson's $r = 0.72$); and BE4max (Pearson's $r = 0.70$), Anc-BE4max (Pearson's $r = 0.87$), and hyA3A-BE4max (Pearson's $r = 0.42$) in HepG2 respectively (Fig. 5d–h). Compared to the Pearson's correlation of 0.64 of BE4max in HEK293T (Fig. 5a), the models of the new base editors and cell types retained good predictive ability with similar and even higher correlation values.

In the model expansion process, we constituted the strategy to adapt the core CAELM model to more types of CBE base editors in various cell types. By training the core model with an additional relatively small set of in situ editing data with desired editors or cell lines, an expanded CAELM model with comparable or even better accuracy could be obtained, which indicated the CAELM prediction compatibility might be universally applicable with the expansion strategy.

## Determination of the relative contribution of DNA accessibility to the editing prediction

Feature importance score is commonly used in assessing how much each input feature contributes to build a model and predict a target variable[29], which provides insight of determining which feature might be most relevant to a target. To determine the approximate contribution of the chromosomal environment condition to the prediction, relative to the contribution of the DNA sequence context, feature importance scores of the sequence context and the DNA accessibility value were obtained respectively from the different models. We used the built-in function get_score() of the XGBoost python package to get the feature importance scores. Instead of learning inputs including sequence information, melting temperature and GC contents[13], we used 2 inputs which were the target sequence context and the DNA accessibility. By the means of one-hot encoding, the target sequence context was transformed into dummy features where each base generated 4 dummy features corresponding to the existence of A, C, G and T, with DNA accessibility as a feature by itself. Therefore, there were $4n + 1$ features in total where n was the number of the bases, and the feature importance score was calculated for each of them. Finally, by dividing the feature importance of the DNA accessibility value by the sum of the importance value of the rest features of the sequence context, we obtained a ratio of 1:6.398, and the expanding models were also analysed, the calculated ratios between the contribution of the DNA accessibility and the DNA context were close to 1:6 respectively (Supplementary Table 4).

To our knowledge, this was the quantitative assessment of the relative importance of the two major factors, DNA sequence and chromatin environment, for determination of the genome editing efficiency of a genomic locus. This result suggested that although the DNA sequence was the major influential factor, the chromatin environment also had substantial influence of the editing efficiency, which could not be ignored.

## Obtaining SNV cell models by sorting of cells from the edited pool

To obtain homogeneous cell models carrying pathogenic SNVs from the batch generated by the automated high-throughput system, ten edited cell pools with relatively higher efficiencies were randomly selected to sort and verify for obtaining homogeneous disease cell models. Then these disease cell models carrying C-to-T SNV could be used for studying and optimizing of adenine base editor (ABE) mediated correction, and development of ABE base editing therapies. The diagram to illustrate this process is showed in Supplementary Fig. 6.

The cells were passaged and expanded for sorting by the fluorescence-activated cell sorting (FACS), the gating strategy for sorting was showed in Supplementary Fig. 7. And the sorted single cells were subsequently proliferated for sequencing verification. Overall, single cells were sorted into 96-well plates, and the average cell survival rate of the 10 pools reached about $16.67 \pm 4.4\%$. Then, the cells were passaged into 24-well plates, in which about $68 \pm 7.24\%$ of cells survived and were subjected to sequencing. The cell lines with 100% C•G-to-T•A transversion at the target SNV sites accounted for $47.30 \pm 5.18\%$ of all the sequenced cells. Finally, 9 disease SNV cell models were successfully constructed (Supplementary Fig. 6b).

Using the obtained cell models, we were able to design gRNAs for ABE mediated correction of the pathogenic SNVs (Supplementary Table 5). After editing, the ABEs showed an average 62% editing efficiency in the nine disease cell models (Supplementary Fig. 6b). For the disease cell models of Gorlin syndrome (VCV000654759) and the sphingolipid activator protein 1 deficiency (VCV000942712), ABE8e[30] showed high correction efficiencies of 98% and 91% respectively. The lowest efficiency of 30% was obtained in the model of hereditary sensory and autonomic neuropathy type IIA (VCV000973155), with a PAM-free ABE8e editor[31]. As shown in the supplementary material. These results suggested that the automated high-throughput system could successfully produce a large scale pathogenic SNV cell models in short time. After subsequent purification, the cell models served as an ideal benchmark for the base editor construction, optimization and analysis. This technique will greatly accelerate the development of genetic therapies for various human genetic diseases.

## Discussion

Human cell lines installed with specific SNVs corresponding to particular diseases are critical models for testing and developing genetic therapies. However, with 37,000 known diseases associated pathogenic SNVs, obtaining even a small portion of these corresponding disease model cell lines is a labor intensive and expensive endeavor. In addition, manual base-editing experiments suffer from human errors and inconsistencies. On the other hand, previously reported machine-learning models for predicting base-editing outcome are based on NGS data of short integrated artificial targeting loci, which may suffer from virus integrating bias and lack of real chromosomal environment of the target sequence[15].

To solve these two problems, we constructed a laboratory automation system for mammalian cell genome editing. Laboratory automation has demonstrated its great capacity by creating a large number of genetic variants in a short time and automated multiplex genome-scale engineering has been realized in *Escherichia coli*, *Corynebacterium glutamicum* and yeast[32–34]. Crucially, automation overcomes repetitive and labor-intensive work and significantly accelerates biological research[35,36]. However, as far as we known, there is still no report on its application for genome editing in mammalian cells, probably due to the difficulty of maintaining the viability and purity of the cell culture. To enable large-scale gene editing applications in mammalian cells, we set out to carefully investigate and modify the protocols to build an automated gene editing platform applicable to mammalian cells. The platform included bioinformatic pipeline and automated equipment, enabling the streamlined generation and analysis of mammalian cell genome editing with high throughput.

In gRNAs plasmid construction, the standard Golden Gate DNA assembly approach was had a reaction volume of 15 μl[37], in order to cut the cost, in our platform acoustic liquid handler (Echo) was used to establish the DNA assembly reaction, which was remarkable for nanoliter-scale and tip-free operation. With a serial of experiment optimization steps, we were able to minimize the reaction system to a total volume of 1 microliter to perform the Golden Gate DNA assembly, which significantly lowered the experimental cost. The analysis of thousands of editing results data often takes a lot of time, in our platform, we established an automated program by Python, which can compare the sequencing results of gRNA plasmid and gene editing efficiency in batches.

We applied this system to 1210 target gene loci for in situ editing in the model mammalian cell line HEK293T. The 1210 gene segments were retrieved with the python package Seqseek. The gRNAs and corresponding PCR verification primers were designed through our bioinformatic pipeline in only 2 h. A total of 1210 gRNA plasmids were constructed in 4 days, with a correct rate of 99%. All gRNA and BE4max plasmids were automatically co-transfected into HEK293T cells within 6 h, and the subsequent medium exchange was completed within 2 h.

The edited cells were harvested for PCR analysis within 8 h, and the editing outcomes were confirmed and analysed by DNA sequencing. Using the high-throughput system, thousands of samples can be automatically edited and analysed within a week.

We compared genome editing efficiencies performed by the automation platform with manual implementation, and found no obvious difference. Furthermore, these data containing thousands of individually edited results from one batch of operations with precise parameters, which provides much higher uniformity and reproducibility than conventional manual operations. By comparisons of manual Vs. high-throughput gene-editing, we found that high-throughput saved both time and cost, particularly the former. This platform not only can provide a foundation for the construction of large-scale libraries of disease cell model, but also facilitate the parallel genetic manipulation of mammalian cell lines. This flexible genome-scale base editing design system combined with the automated platform provides a more convenient and effective method to develop and improve base editors and could be extended to other mammalian cell lines as well.

Based on the highly-uniformed in-situ genome editing data generated by the automation platform, we developed a machine learning model CAELM to predict BE4max behavior in HEK293T cells, and achieved a Pearson's correlation value of 0.64. In contrast to previous machine learning models, the real chromosomal environment of the target sequence was considered in our model, which provided better and more relatively realistic prediction. There are mainly two significant differences between our model and previous models. In the past few years, a variety of different models have been utilized to predict the base editing outcomes of ABEs and CBEs, including CNNs[14] and logistic regression[38]. CAELM employed the XGB Regressor module of the XGBoost, because XGBoost commonly exhibits better performance than deep neural networks when dealing with small datasets. XGBoost uses both L1 and L2 regularization to control over-fitting, and achieves higher performance than some other algorithms. Furthermore, most previous models only used sequence-based information such as single nucleotide, melting temperature and other information as inputs. Instead, we used a different set of inputs, including one-hot encoded target sequence context and the corresponding DNA accessibility value.

Currently, methods were developed using AI tools to learn and predict the CRISPR editing performance[14]. However, the data used by Be-Hive and related models was derived from the editing targets integrated into the chromosome by lentivirus, but not the actual chromosomal loci. Thus, the predicted results might deviate from those from in situ editing, due to the lack of interference from the original chromosomal locus. And the lentivirus also has integration location bias, which distorted the editing data[15] even more. Compared with BE-Hive, CAELM showed a better predictive correlation which trained with in situ data and utilized both sequence and chromatin information as input. In addition, to expand the CAELM prediction compatibility for more universal application, we constituted the strategy to adapt the core CAELM model to more types of CBE base editors in various cell types. A relatively small set of in situ editing data could be used to further train the core model. Theoretically the CAELM might be universally applicable with the expansion strategy.

CAELM displayed good ability to predict base editing efficiencies, with the Pearson's correlation values ranging from 0.42 to 0.87, Pearson's r is among the most prevalent metrics for evaluating the accuracy of models for numerical data[39] and used in many previous literatures[13,14] in studying base editing efficiencies. A r value of exactly 1 or -1 indicates that there is a perfect positive or negative linear dependency between the two features respectively, a r value greater than 0.5 normally implies a strong linear relationship. The machine learning models achieved r values ranging from 0.50 to 0.95 by Song et al.[13], and around 0.7 to 0.9 by Arbab et al.[14], which were higher than the r value of 0.64 we obtained with CAELM. However, it was

reasonable since their models were trained with a greater number of deep sequencing reads compared to ours. But in contrast to previous machine learning models, the real chromosomal environment of the target sequence was considered in our model, which provided better and more relatively realistic prediction. CAELM also provides the feature importance scores of the model inputs and revealed that the contribution of DNA accessibility relative to the target sequence context was close to 1:6 in the prediction.

In conclusion, the automated mammalian cells editing platform is a powerful platform for rapid in situ base editor in a high throughput manner. With the obtain on in situ editing data, we built a machine learning model CAELM considering both the sequence and chromatin environment data, which provided better and more relatively realistic prediction, which was a stepping stone towards a working model that accurately predict the base-editing outcomes in cells from human or other species.

## Methods

### Automated high-throughput platform for gene editing in mammalian cell

The automated editing platform consisted of acoustic liquid handler (Beckman-Echo, USA), plate sealer (Miulab, China), centrifuge (Hettich, Germany), automated thermocycler (Analytik-jena, Germany), automated ClonePix™ system (Genetix, UK), incubator shaker (Infors, Switzerland), liquid handler Beckman i7 (Beckman-Biomek, USA), and Flow cytometer (Beckman, USA). The liquid handling Beckman i7 automated workstation was equipped with an 8-channel independent pipettor, a 96-channel pipettor, a robotic manipulation arm, three Peltier temperature control blocks, and one shaker.

### Plasmids

All gRNA plasmids were constructed by inserting protospacer sequences to replace the *ccdb* coding sequence behind the U6 promoter of the *ccdb*-gRNA plasmid. The *ccdb*-gRNA plasmid was constructed using the Golden Gate method with a standard backbone plasmid (Addgene 51133), the C:G to T:A base editor was pCMV-BE4max-P2A-GFP (Addgene 12099), and the sequences of ABE8e and PAM-free ABE8e are shown in the supplementary materials. The protospacer sequences and gene-specific primers used in this study are listed in Supplementary Data 1, 2, 3.

### Construction of gRNA plasmids

The gRNA plasmids were constructed using the Golden Gate DNA assembly method, and Echo were chosen to assemble Golden Gate reactions by the acoustic droplet ejection. The total reaction system of Golden Gate had a reduced volume of 1 microliter and the protocol details are as the follows. gRNA primers were synthesized in a 384-well source plate. And T4 ligase buffer, *ccdb*-gRNA plasmid, T4 ligase, BSA, BsaI enzyme and ddH$_2$O were placed into another 384-well source plate (Beckman-Echo plate, USA). Firstly, 0.05 µl T4 ligase buffer, 0.35 µl ddH$_2$O, 0.05 µl forward and 0.05 µl reverse primers were transferred from the 384-well source plate to the 96-well PCR plate (Axygen, USA) by Echo acoustic droplet ejection, and then sealed by the plate sealer (180 °C, 4 s). The plate was centrifuged (5000 × *g*, 3 min) and annealed (95 °C, 5 min) in the thermocycler to obtain dsDNA. Secondly, 50 ng *ccdb* plasmid, 0.1 µl T4 ligase, 0.1 µl BSA and 0.06 µl BsaI enzyme were transferred from the 384-well source plate to a dsDNA 96-well PCR plate via the Echo acoustic droplet ejection, and then filled with ddH$_2$O to a 1 µl reaction system by Echo. The plate was then sealed (180 °C, 4 s), and centrifuged (5000 × *g*, 3 min). The reaction was carried out in a thermocycler using the following program: 37 °C for 3 min, 16 °C for 4 min, repeat 1 − 2 for 25 cycles, 50 °C for 4 min, 80 °C for 5 min, 4 °C hold. Third, DH5α competent cells (Cwbio, China) were prepared with an *E. coli* Transformation Kit & Buffer Set (Zymo Research, CA). The DH5α competent cells were added to the 96-well PCR plate carrying the

Golden Gate reaction products on a Peltier block by 8-channel liquid handler, and incubated at 0 °C for 5 min. Then the transformants were transferred to sharp bottom 96-well plates (Corning, USA) by the 96-channel liquid handler. The cell suspensions were spread onto LB agar plates with 100 µg/mL of ampicillin by the ClonePix™ system, followed by overnight incubation at 37 °C. Then, single colonies were picked and cultured in the high-throughput incubator shaker at 37 °C, 800 rpm. The constructed gRNA plasmids were verified by Sanger sequencing, and the correct plasmids were extracted using the MagBeads Plasmid DNA Extraction Kit (Biomiga, China) according to the manufacturer's protocol by the Beckman i7 liquid handler.

### Mammalian cell culture and transfection

HEK293T and HepG2 cells were obtained from the American Type Culture Collection (ATCC), and cultured in Dulbecco's modified Eagle's medium (DMEM) (Gibco, Thermo Fisher, CN) supplemented with 10% fetal bovine serum (FBS) (Gibco, Thermo Fisher, CN) at 37 °C in a 5% CO$_2$ incubator. The liquid handler transfection protocol was as the follows. Cells were seeded into 96-well poly-d-lysine-coated plates (Corning, USA) to an approximated $1 \times 10^4$ cells per well by the 96-channel liquid handler. Between 16 and 24 h post seeding, the cells were transfected at approximately 60% confluency with 0.75 µl of PEI (1 mg/ml, Yeasen, China) plus 160 ng of base editor plasmid and 80 ng of sgRNA plasmid per well. Four to six hours after cell transfection, the transfection media was changed with 100 µl of DMEM containing 10% FBS per well. On the third day, cells were cultured in medium containing 4 µg/mL puromycin (Solarbio, China) and then cultured in medium containing 2 µg/mL puromycin once.

### Genomic DNA extraction and sequencing

Transfected cells were collected by the Beckman i7 liquid handler, centrifuged and transferred to the thermocycler. The protocol details were as follows. Cells were washed with 100 µl 1× PBS solution (Gibco, USA) and then resuspended in 100 µl PBS by repeatedly pipetting at eight points along the circle of wells. Then the supernatant was removed by centrifugation, and genomic DNA was extracted by the addition of 10 µl freshly prepared lysis buffer (10 mM Tris-HCl pH 8.0, 50 mM KCl, 0.05% Nonidet P-40, 100 µg/ml Protainase K (Solarbio, China)), which was directly loaded into each well of the tissue culture plate by an 8-channel liquid handler. The genomic DNA/lysis buffer mixture was incubated at 65 °C for 10 min and 98 °C for 2 min in a thermocycler.

PCR products for sequencing were obtained by Echo, Beckman i7 liquid handler, centrifuge, plate sealer and thermocycler. The protocol for performing PCR in the 30 µl volume was as follows. First, 1.5 µl forward primers and reverse primers for target DNA amplification were transferred from the 384-well source plate to the 96-well PCR plate by Echo, respectively. Then, 15 µl 2×ES Taq MasterMix (Cwbio, China) and 7 µl ddH$_2$O were consecutively added by the 8-channel liquid handler, and 3 µl of cell lysate was added as a PCR template by the 96-channel liquid handler. The plates were then sealed (180 °C, 4 s), centrifuged (5000 × *g*, 3 min). The PCR reactions were carried out by the thermocycler with the following program, 94 °C for 3 min, then 30 cycles of (94 °C for 30 s, 60 °C for 30 s, 72 °C for 30 s), followed by a final 72 °C extension for 2 min, and hold at 4 °C. The primers used for genome DNA amplification are listed in Supplementary Data 2. The efficiencies of gene editing by the automated platform were finally confirmed by Sanger sequencing and quantified by EditR[21].

### Obtaining the chromatin accessibility information

Chromatin accessibility is mainly determined by the occupancy and topology of nucleosomes and chromatin-binding factors[32]. The main method for detecting chromatin accessibility is the foot printing method with DNase I, which mainly cleaves DNA regions that are not occupied by nucleosomes, and the protected regions are determined

by sequencing of undigested DNA fragments. The chromatin accessibility data was retrieved from the ENCODE. For each target site, the protospacer adjacent motif (PAM) plus protospacer sequence were aligned to the GRCh37/hg19 human reference genome, and the average value of the DNase I HS Density Signal over the protospacer sequence were retrieved from ENCODE with UCSC Accession (Supplementary Data 9–10).

### Prediction of the in situ base editing efficiency

One-hot encoding generated features from the target sequences along with the corresponding DNA accessibility value, which was retrieved from ENCODE, and UCSC Genome Browser database track on the GRCh37/hg19 human assembly, were fed into a XGB Regressor model to predict the BE4max editing efficiencies. We used a simple train/test split from the entire dataset consisting of target sequences at an 80/20 ratio, and the GridSearchCV method was used to split the training set with 5-fold cross-validation to gain the optimal hyperparameters. The min-max approach was used to scale the DNA accessibility value and the target variable (BE4max editing efficiency) into the range of [0,1]. In order to evaluate the contribution of the 20-mers of the protospacer and the DNA accessibility to the base editing efficiency, the ratio of the contribution of the DNA sequence context and the DNA accessibility was determined by calculating the ratio between the corresponding feature importance scores obtained from the trained model. The built-in XGBoost function get_score() was used to evaluate the importance of all features including the sequence context and the DNA accessibility value. The relative importance of the DNA accessibility was calculated by dividing the feature importance of DNA accessibility value by the sum of the features importance value of the sequence context.

### Isolation of the disease-associated SNV cell models and sequence verification

To obtain the pure disease-associated SNV cell models, single cells were sorted by FACS. The edited cell pools from 96-well poly-d-lysine-coated plates were passaged and expanded into 24-well poly-d-lysine-coated plates. After 2 or 3 days, cells were harvested, and single cells were sorted according to the GFP fluorescence generated due to the reporter gene carried by the BE4max plasmid. Single cells were sorted into 96-well poly-d-lysine-coated plates containing 100 μl of DMEM with 10% FBS per well. Sorted plates were incubated for 14 days until well-characterized colonies were visible, with periodic media changes performed as necessary, and then passaged and expanded into 12-well poly-d-lysine-coated plates after another 2 or 3 days. The single cells of disease models isolated by FACS sorting were finally confirmed by PCR followed sanger sequencing and quantified by EditR.

### Statistical & reproducibility

Samples were assigned randomly during data analysis and model building. To determine the significance of the base editing efficiency of Manual Vs. high-throughput gene editing, we used the two-tailed Student's *t*-test for comparison of the two operations. All experiments were performed for three independent biological replicates. Statistical analyses were conducted using Graphpad Prism. *P* values less than 0.05 were considered statistically significant.

### Reporting summary

Further information on research design is available in the Nature Portfolio Reporting Summary linked to this article.

## Data availability

There is no restriction on data associated with this study. The Sanger sequencing data from this study have been submitted to the NCBI Sequence Read Archive under accession number PRJNA885770. The data sets used in this study are provided as Supplementary Data 1-10 and Supplementary Table 1–5. The plasmids for expression of gRNAs and BEs are listed in the Supplementary Information file (Supplementary Fig. 8–13). These Python scripts and the prediction models are available on GitHub (https://github.com/YQLiCAS/BE4max[40]) for public research use, and the DOI for it is https://doi.org/10.6084/m9.figshare.21547509.v1. Source data are provided with this paper.

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

## Acknowledgements

This research was financially supported by the National Key R&D Program of China (2018YFA0902900, 2018YFA0901300), National Natural Science Foundation of China (31861143019, 32171449, 32001041, 81972700, 61827819), the Tianjin Synthetic Biotechnology Innovation Capacity Improvement Project (TSBICIP-KJGG-017, TSBICIP-PTJS-003), Youth Innovation Promotion Association CAS and Tianjin Natural Science Foundation (20JCYBJC00310).

## Author contributions

X.L.Z., C.B. and M.W. designed the research, analyzed data, and wrote the manuscript. S.L and J.A. designed the research, performed experiments, analysed data, and wrote the manuscript. Y.L., X.G.Z., and Y.Y. analysed data and wrote the manuscript. D.Z., L.W., and Y.S. performed experiments.

## Competing interests

A provisional patent has been submitted in part entailing the reported approach. The patent author contains M.W., J.A., X.L.Z., C.B., S.L., Y.L., the institution is "Tianjin Institute of Industrial Biotechnology, Chinese Academy of Sciences, Tianjin, China", and the patent application number: CN202210360423.9. The status is under review. The title is a high-throughput method of construction of mammalian cell disease models, specific aspect of manuscript covered in patent application: Development of an automated platform for genome editing in mammalian cells in Results of manuscript. X.G.Z., Y.Y., D.Z., L.W., and Y.S. declare no potential conflict of interest.
