## [Peer Review File · Nature Communications]

Reviewers' Comments:

Reviewer #1:

Remarks to the Author:

The paper by Li et al. developed an automated platform to design and build in situ genomic editing at 1201 gene loci within the model mammalian cell line HEK293T. Next, the authors developed an AI learning model based on sequence information and a chromatin accessibility database to predict the efficiency of each base editing target site. Considering the existing difficulties in handling such cells in an automated platform while keeping cell culturing protocols at a certain purity level, the authors present what seems to be the first automation approach to design, fabricate and test gene editing in mammalian cells in a high-throughput, automated way. The work has scientific relevance and contains significant results in the field. However, at its present stage, I suggest it not be published.

In my opinion, the authors should heavily change figures, revise the important points and concerns highlighted here, and perform a thorough English revision of the entire manuscript before it can be considered for publication at Nature Communications.

1) I have added a few examples of revisions to be considered, starting from the Abstract and Introduction sections. Here are some points to help the author address the problems of the manuscript:

-In the Abstract, please remove the sentence 'which was uploaded to GitHub for public usage.' Then, add the GitHub link to the data availability section at the end of the manuscript.

The language is not appropriate in the abstract: 'The work will hopefully accelerate the development of BE-based genetic therapies.' You can re-do the phrase to replace 'hopefully' with something like 'It is expected.'

-In the introduction, please describe the actual advantage of using a 'real' chromosomal environment of the target sequence in machine learning models, as compared to other models recently published and cited by you. Please provide the most recent example (or citations) of a model that lacked a realistic parameter, such as the one implemented in your work.

-In the introduction, please add one or more citations for the following phrase: 'Furthermore, for large scale samples, manual operations are not only time-consuming, but also with high error rate, less consistency and high cost.' Else, present this sentence as part of the results you are giving, as per one of my suggestions in the Figures (in the comments below).

In the introduction, the connection between automation methods and the possibility of accessing the local chromatin information is not clear in this phrase. Please rephrase it and clarify. 'true automatic high-throughput method could provide a feasible solution, due to the possibility to access the local chromatin information'

-In the introduction and throughout the manuscript (if applicable), be careful with using the terms 'fully automated'. For example, I would imagine that a fully automated platform would not have humans intervene in any step whatsoever. Is this the case? If not, please indicate the steps where there is some or minimum human innervation in Fig.1.

-In the introduction, 'guide RNA (gRNA) design to the analysis of the editing results, which comprehensively characterizes the relationships of the in situ base editing outcomes with the sequence and chromatin environment for BEs,' then 'the context-sequence information and a chromatin accessibility parameter.' Please clarify how this manuscript has characterized the relationship between BE outcomes and chromatin environment (chromatin parameter?). Is this something done in the past or invented in this paper? If this has been described later in the Results section, please add at least the logic behind this connection for broader audiences.

-In the introduction, 'Based on the automated mammalian cell editing platform, genome-wide editing can be implemented in a scalable manner using a standardized workflow,' add the name

and a citation for the standardized workflow.

-In the Results, please rephrase and break the following sentence for a better flow and cohesion: 'In this work, with the automated high-throughput system, we obtained a large-scale dataset of in situ editing of mammalian cell with high uniformity, which could not be accessed by manual operation, we could construct an AI learning model employing both the context-sequence information and chromatin accessibility parameter to predict the in situ base editing results.' The words 'could construct' add confusion to whether a new model has been created or not in this work.

Next, please describe which type of model has been exclusively created in this work and how it differs from other literature models and others used in the study. For example, does it have a name and Acronym for better reference inside the manuscript? Finally, given the importance of the new method, I suggest creating a dedicated section for it (in Methods) to describe the new method and add comparisons.

2) Here, I am adding some major English mistakes found in the manuscript, to name a few, which I believe could help the authors in the revision process:

-In the abstract. This sentence could be broken into two: 'It is necessary to construct cell disease models for developing and optimizing BEs, for large-scale studies, manual operations are not only time-consuming, but also result in high costs and increase the error rate.'

-In the abstract. This sentence is too long and hard to follow: 'Based on the high-uniformity in-situ genome editing data uniquely obtained by the automatic high-throughput system, we developed a new machine learning model to predict BE4max behavior, this is the first time that the real chromosomal environment of the target sequence was considered in machine learning models, which provided relatively realistic and better prediction.'

-Bad grammar used in the caption of Fig. 4: 'The Models of predict for base editing'

-Not scientific, formal English: 'Previous research always used the methods' Avoid the use of 'always,' 'never,' etc. In this manuscript, you may not know the exceptions in the whole literature just yet.

3) Here are the most critical edits of the manuscript related to the figures:

-In Fig. 1:

-This figure seems too crowded in its present state, and it could be divided into two figures. First, a general picture of the workflow with the cartoons, the 'a', 'b,' 'c,' and 'd' section names, and their infrastructure.

-Second, the sections and their associated tasks. The latter could be expanded to present the file formats and type of data exchanged among sections and sub-sections, indicating how the connections can be made.

-Then, for each section (a, b, c, and d), for an actual case study (e.g., one of the targets) that went through the entire workflow, please add an example figure (that could go to supplementary information) that can demonstrate file formats, example input/output files.

-Some of the images/cartoons seem too pixelated. Please make sure the resolution of the cartoons/illustrations and other images follow the journal's guidelines.

-In Fig. 2:

-This figure should go to supplementary information, and it should be used to complement the comparisons of Manual Vs. high-throughput gene-editing methods. For example, it would be interesting to add the time taken if done manually, with an intelligent way to compare the most time-limiting steps of both workflows.

-Then, I would add a comparison of costs to determine whether the high-throughput method can also be advantageous in economic terms. Perhaps a table for cost comparisons would make more sense to present the differences if all data in the figure becomes crowded.

-The results and conclusions of this comparison could be added back to the main manuscript in the respective sections and, depending on their impact, be presented as a table (or a bar plot) of the comparisons (time and costs).

-In Fig. 3:

-I would add another panel in this figure to demonstrate the formula/equations (then references in Methods) to explain how base editing efficiencies are calculated in the first place.

-In Panel-A, you could add an Inset in the plot to show the results of the comparison between the base editing efficiency of Manual Vs. high-throughput gene editing, aiming to show that results at high-throughput gene editing are higher than Manual operation at 32 genomic loci. Then, present a statistical method and calculation (KS-test?) to determine whether the difference between Manual Vs. high-throughput gene editing is statistically different.

-The tables in sections 'b' and 'e' could be moved to supplementary information as they are redundant to results presented in sections 'a' and 'd'.

-This figure benefits from having genuine sequence fragments and base editing examples to demonstrate the categories found (No Editing, <5%, 6-10%, 11-49%, >50%, etc.).

-We should be clear about the difference between the number of target sites changed as the base location within an endogenous genomic locus. Again, another cartoon figure could be added to this figure to represent the strategy behind the base edits aimed.

-In Fig. 4:

-In Panel-ZERO (first one), can you have error bars (based on biological replicates) for the actual base editing efficiency for better comparison? Are the statistical methods able to generate the same for better comparison?

-In the same Panel-ZERO (the new one), I would add some examples of base editing using the BE4max and A3A-nCas9 methods to demonstrate their molecular mechanisms. A broader audience could benefit from having a better understanding of them.

-In panel-C, it is not clear what result or comparison should be made or expected. For example, some pair-wise comparisons could be between the actual and other model results indicated inside the plot to represent the method's performance developed in this work.

-In panel-B, please define N20s in the figure legend.

-In Fig. 5:

-Correct the position of the red rectangles where base locations are highlighted; some of them are misaligned and may create confusion.

-What is the takeaway message from this figure? Can you add the results you wish your audience to observe and compare, perhaps next to the red rectangles?

4) In the Results section, please address the following points:

The abstract presents the following sentence: 'And the chromatin accessibility parameter was calculated to take up 1/6 importance relative to the context-sequence.' I could not find any figure or table that could demonstrate this result itself or its relative importance to other parameters.

-First, in Results, the Pearson correlation is missing a citation. Second, the following phrase is not accurate and potentially statistically wrong: 'A reasonable correlation was found between the experimental and predicted values (Pearson's correlation coefficient $r = 0.64$), which suggests a decent prediction accuracy' What is the best way to describe these results in a statistical sense? What is a decent accuracy in a statistical sense or given some benchmark?

-How were 'feature importance scores' obtained from the model? Is this a method that has been published or a new method developed in this study?

-Next, in the following result, 'To our best knowledge, this is the first quantitative assessment of the relative importance of the two major factors, DNA sequence and chromatin environment, for determination the genome editing efficiency of a genomic locus. This result suggested that although the DNA sequence was the major influential factor, but the chromatin environment also determined the editing efficiency and could not be ignored.' If the method used to determine the influence of each factor in shaping the editing efficiency was created in this study, please add at

least one more independent method (preferably known) to validate these results.

5) For data availability, there could be repositories where a link for download could be provided. In any case, please make sure it follows the journal's guidelines.

6) Editorial board questions

-What are the noteworthy results?

R. I guess so; I can only say for sure after a second revision with my concerns being addressed.

-Will the work be of significance to the field and related fields? How does it compare to the established literature? If the work is not original, please provide relevant references.

R. Yes.

-Does the work support the conclusions and claims, or is additional evidence needed?

R. Probably. I could not find information about technical or biological replicates for the results found.

-Are there any flaws in the data analysis, interpretation, and conclusions? Do these prohibit the publication or require revision?

R. Fairly. The work could benefit from having a deeper statistical analysis to support the results and conclusions, particularly when claiming that one method is more accurate than the other.

-Is the methodology sound? Does the work meet the expected standards in your field?

R. Fairly. Some methodologies need to be better described.

-Is there enough detail provided in the methods for the work to be reproduced?

R. Fairly. In my opinion, as suggested, the manuscript will benefit by having data types and formats shown in new supplementary figures that demonstrate template files.

Reviewer #2:

Remarks to the Author:

This paper studies in situ base editing efficiency, and provides a dataset of base editing efficiency measured at 1134 genomic loci in the native chromatin and epigenomic contexts. While the trained machine learning model is limited in scope, and its performance does not obsolete BE-Hive, I expect that the paper will be of interest to scientists interested in the effect of chromatin context on base editing efficiency.

The authors take a reasonable and standard approach in data processing and training the machine learning model. The paper's clarity could be improved by specifying the Sanger sequencing read count in the methods. The authors describe proper cross-validation for hyperparameter selection, and test set evaluation. However, the paper's clarity could be improved by reworking the caption for figure 4, and clarifying in the main text that the reported performance ($r=0.64$) is on the held-out 20%. The paper could be strengthened by performing full k-fold test set evaluation: if the test set is 20% of the full dataset, train and test 5 times on each non-overlapping split and report 5 test-set performances for the trained model and BE-Hive. This would enable readers to better evaluate the model's improvements over BE-Hive.

The machine learning model's contributions may be limited in scope, as it is trained on only one base editor (BE4Max) in one cell-type (HEK293T). As such, it is unlikely that the model will achieve broad use by the base editing community. This contribution could potentially be strengthened by proposing a machine learning approach to extrapolate from learned chromatin accessibility dependencies to other base editors or cell-types; such approaches could be evaluated using the small dataset of 30 genomic loci reported to be edited by both APOBEC3a-nCas9 and BE4Max in the paper.

The paper's larger contribution may be its systematic study of the contribution of chromatin context to base editing efficiency. The analysis supporting the conclusion of 1/6 relative

importance appears sound to me. However, the paper is lacking any mention or discussion of prior work that study the relevance of chromatin context to Cas9 editing. The paper would be strengthened if such comparisons or discussion were included.

Reviewer #3:

Remarks to the Author:

The authors present here an automated genetic editing platform combined with machine learning approaches to predict editing efficiency. The genetic editor involves a base editor, capable of changing a single base pair without the need to induce double-stranded DNA breaks. These types of genetic changes are of interest in mammalian cells because single nucleotide variations are known to be involved in a large fraction of human diseases. Hence, cell disease model cells are desirable for research purposes.

In my opinion, most of the value of this work lays in the production of the automated pipeline. Automating molecular biology is an arduous process and the claim of a fully automated process is very interesting. However, I have my qualms that the authors have truly achieved a "fully automated process" in the sense that no human intervention is needed from design to prediction, given the equipment description. It looks more like an automated pipeline in which humans transport cells and reagents from phase to phase. Merit worthy and interesting, but less impressive.

The machine learning approach, using an existing single standard method (XGB) to leverage the large amounts of data generated (~ 1000 genetically edited cells) to predict editing efficiency is, in comparison, a rather less imposing effort. The results are better than a previously published neural net ML method (BE-Hive), but not by much (0.64 vs 0.52 Pearson correlation coefficient). The only real novelty is the use of the chromatin accessibility information as part of the input vector for the machine learning algorithm, despite claims to the contrary ("we developed a new machine learning model..."). The estimate of its relative importance in predicting the gene editing efficiency is novel and somewhat valuable (estimated to be ~1/6 of all importance scores).

In summary, I believe that the most valuable part of this work relies in the automated pipeline able to produce thousands of genetically edited model cells. I don't, however, work on mammalian cells or health, so I feel I am not in a good position to determine the overall value of the paper to the field. I will leave that to the experts.

Finally, I would say that the paper is hard to read for non-specialists and needs some improvement in its grammar to be easily digested.

Reviewer #1

(Remarks to the Author):

The paper by Li et al. developed an automated platform to design and build in situ genomic editing at 1201 gene loci within the model mammalian cell line HEK293T. Next, the authors developed an AI learning model based on sequence information and a chromatin accessibility database to predict the efficiency of each base editing target site. Considering the existing difficulties in handling such cells in an automated platform while keeping cell culturing protocols at a certain purity level, the authors present what seems to be the first automation approach to design, fabricate and test gene editing in mammalian cells in a high-throughput, automated way. The work has scientific relevance and contains significant results in the field. However, at its present stage, I suggest it not be published.

In my opinion, the authors should heavily change figures, revise the important points and concerns highlighted here, and perform a thorough English revision of the entire manuscript before it can be considered for publication at Nature Communications.

Response: Thank you very much for giving us the opportunity to revise the manuscript. We appreciate your constructive comments and suggestions and have revised our manuscript carefully and thoughtfully based on each point raised. The changes were highlighted with blue color in the revised manuscript in responding to the reviewers' questions and suggestions. We have performed a thorough English revision of the entire manuscript and remade some of the figures. We hope that the revised manuscript has answered your questions and improved following the suggestions. Thanks again for the help.

1. I have added a few examples of revisions to be considered, starting from the Abstract and Introduction sections. Here are some points to help the author address the problems of the manuscript:

(1) In the Abstract, please remove the sentence 'which was uploaded to GitHub for public usage.' Then, add the GitHub link to the data availability section at the end of the manuscript.

The language is not appropriate in the abstract: 'The work will hopefully accelerate the development of BE-based genetic therapies.' You can re-do the phrase to replace 'hopefully' with something like 'It is expected.'

-In the introduction, please describe the actual advantage of using a 'real' chromosomal environment of the target sequence in machine learning models, as compared to other models recently published and cited by you. Please provide the most recent example (or citations) of a model that lacked a realistic parameter, such as the one implemented in your work.

Response: Thank you for these constructive comments. We have added the GitHub link to the data availability section at the end of the manuscript (Data availability), the abstract was re-written accordingly. Critical citations on the importance of chromosomal environment for machine learning models were provided.

The phrase in the introduction was modified to the following.

“Previous researches have shown a strong correlation between the performance of nuclease and the chromatin accessibility properties^{14, 16}. Kristopher et al. demonstrated that gene editing was more efficient in euchromatin than in heterochromatin¹⁶. Large-scale genetic screens in human cell lines indicated that highly active sgRNAs for Cas9 and dCas9 were found in regions of low

nucleosome occupancy, and the nucleosomes directly impeded Cas9 binding and cleavage in vitro¹⁷. We previously found that pioneer factor, such as Vp64, improved CRISPR-based C-to-G and C-to-T base editing by changing local chromatin environment¹⁸. However, current studies on deep learning employed the editing data from lentiviral integrated target sequences, while the real chromosomal environment of the target sequence was ignored^{13, 14}. One of the reasons is that it is difficult to obtain a large set of editing data from endogenous target sites.”

(2) In the introduction, please add one or more citations for the following phrase: 'Furthermore, for large scale samples, manual operations are not only time-consuming, but also with high error rate, less consistency and high cost.' Else, present this sentence as part of the results you are giving, as per one of my suggestions in the Figures (in the comments below).

Response: Thank you for your suggestion. As suggested, the relevant citations were added in the article, please see Page 3, Line 88. Meanwhile, we add a table for comparisons of Manual Vs. high-throughput gene-editing cost. We found that a large difference in time consumption per step between manual and high-throughput. Furthermore, the more the samples, the more time is saved by high throughput platform. In terms of cost per sample, the high-throughput platform is lower than that of manual operation in general, but may vary significantly depending on high education personnel costs in different country. The detailed comparisons were supplied in supplementary Fig. 3.

(3) In the introduction, the connection between automation methods and the possibility of accessing the local chromatin information is not clear in this phrase. Please rephrase it and clarify. 'true automatic high-throughput method could provide a feasible solution, due to the possibility to access the local chromatin information'

Response: Thank you for your suggestion. As suggested, we have clarified the connection between automation methods and the possibility of accessing the local chromatin information. The description has been changed to the following content in the revised manuscript.

“An automatic platform would make it possible to get large-scale editing dataset of endogenous targets. And with the large-scale in situ editing data and sequence information, combined with local chromatin accessibility, a machine learning model with in situ data might be able to better predict the actual base editing efficiency.” And the DNaseI HS Density Signal data represented the chromatin accessibility of target sites as previously described, which was employed as the input information for the predict-learning model building.”

(4) In the introduction and throughout the manuscript (if applicable), be careful with using the terms 'fully automated'. For example, I would imagine that a fully automated platform would not have humans intervene in any step whatsoever. Is this the case? If not, please indicate the steps where there is some or minimum human innervation in Fig.1.

Response: Thank you for your suggestion. In this case, some steps in this work still needs human intervention. The phrase “fully automated” in this manuscript was changed to “automated”. Meanwhile, in order to illustrate the automation process more clearly, we modified Fig. 1, and supplied the detail workflow of each module in supplementary Fig. 1. The inputs, outputs, and

human intervention steps in modules are demonstrated in Supplementary Fig. 2.

(5) In the introduction, 'guide RNA (gRNA) design to the analysis of the editing results, which comprehensively characterizes the relationships of the *in situ* base editing outcomes with the sequence and chromatin environment for BEs,' then 'the context-sequence information and a chromatin accessibility parameter.' Please clarify how this manuscript has characterized the relationship between BE outcomes and chromatin environment (chromatin parameter?). Is this something done in the past or invented in this paper? If this has been described later in the Results section, please add at least the logic behind this connection for broader audiences.

Response: Thank you for these constructive comments. We have added following content to clarify the current understanding of the relationship between BE outcomes and chromatin environment in introduction. "Previous researches have shown a strong correlation between the performance of nuclease and the chromatin accessibility properties^{14, 16}. Kristopher et al. demonstrated that gene editing was more efficient in euchromatin than in heterochromatin¹⁶. Large-scale genetic screens in human cell lines indicated that highly active sgRNAs for Cas9 and dCas9 were found in regions of low nucleosome occupancy, and the nucleosomes directly impeded Cas9 binding and cleavage *in vitro*¹⁷. We previously found that pioneer factor, such as Vp64, improved CRISPR-based C-to-G and C-to-T base editing by changing local chromatin environment¹⁸. However, current studies on deep learning employed the editing data from lentiviral integrated target sequences, while the real chromosomal environment of the target sequence was ignored^{13, 14}. One of the reasons is that it is difficult to obtain a large set of editing data from endogenous target sites."

(6) In the introduction, 'Based on the automated mammalian cell editing platform, genome-wide editing can be implemented in a scalable manner using a standardized workflow,' add the name and a citation for the standardized workflow.

Response: Thank you for your suggestion. We originally mean that the workflow described in this manuscript can be the standardized workflow to facilitate wide application of the automated mammalian cell editing platform. Now, because we rewrote the introduction, streamlined the language and found that this sentence was not very precise, so we removed it.

(7) In the Results, please rephrase and break the following sentence for a better flow and cohesion: 'In this work, with the automated high-throughput system, we obtained a large-scale dataset of *in situ* editing of mammalian cell with high uniformity, which could not be accessed by manual operation, we could construct an AI learning model employing both the context-sequence information and chromatin accessibility parameter to predict the *in situ* base editing results.' The words 'could construct' add confusion to whether a new model has been created or not in this work.

Response: Thank you for these constructive comments. We have rephrased the confusing sentence in the revised version.

"In this work, with the automated high-throughput genome editing system, we obtained a large-scale data of *in situ* editing in mammalian cell. The datasets have high uniformity and repeatability, which could not be accessed easily by manual operation.", please see, Page 8, Line 217-220.

“Based on the above 1134 valid editing results, with the context-sequence information and chromatin accessibility value of each endogenous target sites, we constructed machine learning models to predict the in situ base editing results of human cell lines. CAELM was designated as the Acronym for the High-Throughput Automatic Learning Model in this work.”, please see Page 8, Line 228-232.

(8) Next, please describe which type of model has been exclusively created in this work and how it differs from other literature models and others used in the study. For example, does it have a name and Acronym for better reference inside the manuscript? Finally, given the importance of the new method, I suggest creating a dedicated section for it (in Methods) to describe the new method and add comparisons.

Response: Thank you for these constructive comments. We have described the difference between our model and other literature models in the revised version, the method for our models building was created in the Methods section as a dedicated section. For better reference, **CAELM** is designated as the Acronym for the **Chromatin Accessibility Enabled Learning Model** in this work. “There are mainly two significant differences between our model and previous models, in the past few years, a variety of different models have been utilized to predict the base editing outcomes of ABEs and CBEs, including CNNs¹⁴ and logistic regression³⁸, CAELM employed the XGB Regressor module of the XGBoost, because XGBoost commonly exhibits better performance than deep neural networks when dealing with small datasets. XGBoost uses both L1 and L2 regularization to control over-fitting, and achieves higher performance than some other algorithms. Furthermore, most previous models only used sequence-based information such as single nucleotide, melting temperature and other information as inputs. Instead, we used a different set of inputs, including one-hot encoded target sequence context and the corresponding DNA accessibility value.”, please see Page15, Line 419-430.

2. Here, I am adding some major English mistakes found in the manuscript, to name a few, which I believe could help the authors in the revision process:

(1) In the abstract. This sentence could be broken into two: 'It is necessary to construct cell disease models for developing and optimizing BEs, for large-scale studies, manual operations are not only time-consuming, but also result in high costs and increase the error rate.'

Response: Thank you for these constructive comments. We have rephrased the confusing sentence. “A great number of cell disease models with pathogenic SNVs are needed for the development of genome editing based therapeutics or broadly basic scientific research. However, the generation of traditional cell disease models is heavily dependent on large-scale manual operations, which is not only time-consuming, but also costly and error-prone.”

(2) In the abstract. This sentence is too long and hard to follow: 'Based on the high-uniformity in-situ genome editing data uniquely obtained by the automatic high-throughput system, we developed a new machine learning model to predict BE4max behavior, this is the first time that the real chromosomal environment of the target sequence was considered in machine learning models, which provided relatively realistic and better prediction.'

-Bad grammar used in the caption of Fig. 4: 'The Models of predict for base editing'

-Not scientific, formal English: 'Previous research always used the methods' Avoid the use of

'always,' 'never,' etc. In this manuscript, you may not know the exceptions in the whole literature just yet.

Response: Thank you for these constructive comments. We have rephrased the confusing sentence. We carefully checked the manuscript and corrected the mentioned errors.

“Based on the *in situ* genome editing data obtained by the automatic high-throughput platform, we developed a new Chromatin Accessibility Enabled Learning Model (CAELM), to predict the performance of cytosine base editors (CBEs). For the first time, the real chromosomal environment of the target sequences was considered in models building, which provided better prediction based on the realistic data.”

The caption of Fig. 4 was modified to “Fig. 5. The performance of the CAELM Models for prediction base editing outcome.”

And we carefully polished our English writing throughout the manuscript.

3. Here are the most critical edits of the manuscript related to the figures:

(1) In Fig. 1:

-This figure seems too crowded in its present state, and it could be divided into two figures. First, a general picture of the workflow with the cartoons, the 'a', 'b,' 'c,' and 'd' section names, and their infrastructure.

-Second, the sections and their associated tasks. The latter could be expanded to present the file formats and type of data exchanged among sections and sub-sections, indicating how the connections can be made.

-Then, for each section (a, b, c, and d), for an actual case study (e.g., one of the targets) that went through the entire workflow, please add an example figure (that could go to supplementary information) that can demonstrate file formats, example input/output files.

-Some of the images/cartoons seem too pixelated. Please make sure the resolution of the cartoons/illustrations and other images follow the journal's guidelines.

Response: Thank you for your very constructive suggestion. As suggested, we divided Fig. 1 into three figures and improved the resolution. The revised Fig. 1 shows overview of automated high-throughput platform of gene editing in mammalian cells. The revised Fig. 2 shows schematic overview for the automated gRNA plasmids construction workflow. The new Fig. 3 shows overview of the workflow of the automated gene editing process. In addition, the new figures contain actual case study and the input/output files, the detailed files formats and results is provided in supplementary Data 4-8. The detailed workflow of each module is illustrated in supplementary Fig. 1. The inputs, outputs, and human intervention steps in modules of the automated mammalian cells gene editing platform are showed in Supplementary Fig. 2. And the details of CSV output files containing sequencing comparison results are add to the supplementary information Data 1-10.

(2) In Fig. 2:

-This figure should go to supplementary information, and it should be used to complement the comparisons of Manual Vs. high-throughput gene-editing methods. For example, it would be interesting to add the time taken if done manually, with an intelligent way to compare the most time-limiting steps of both workflows.

-Then, I would add a comparison of costs to determine whether the high-throughput method can also be advantageous in economic terms. Perhaps a table for cost comparisons would make more

sense to present the differences if all data in the figure becomes crowded.

-The results and conclusions of this comparison could be added back to the main manuscript in the respective sections and, depending on their impact, be presented as a table (or a bar plot) of the comparisons (time and costs).

Response: Thanks for the constructive suggestion. As suggested, we moved Fig. 2 to the supplementary information, and added the comparison of the time and cost of manual versus high-throughput. Furthermore, we compared the details of each step, in terms of time and expense. The results showed that high-throughput platform was superior to manual operation, dealing with multiple samples in terms of time. And with the increase of sample number, the advantage of the automatic platform may be more obvious. In terms of cost per sample, the high-throughput platform is lower than that of manual operation in general, but may vary significantly depending on high education personnel costs in different country. The detailed comparisons were supplied in supplementary Fig. 3. And the results of this comparison were added into the article, “In summary, we developed an automated platform for high-throughput in situ gene editing of mammalian cells, allowing the parallel processing of a considerable number of samples, which simplified the repetitive and labour-intensive manual laboratory work and provided high-quality data for further machine learning. Detailed workflow of each module, the inputs, outputs, and human intervention steps in each module of the automated gene editing in mammalian cells are shown in supplementary Fig. 1 and Fig. 2. By comparison, we found that high-throughput was superior to manual operation, both in terms of time and cost. With the increase of sample size, the advantage of automated high-throughput platform will be more significant. The detailed comparisons are supplied in supplementary Fig. 3.”, please see Page 6-7, Line 171-180.

(3) In Fig. 3:

-I would add another panel in this figure to demonstrate the formula/equations (then references in Methods) to explain how base editing efficiencies are calculated in the first place.

-In Panel-A, you could add an Inset in the plot to show the results of the comparison between the base editing efficiency of Manual Vs. high-throughput gene editing, aiming to show that results at high-throughput gene editing are higher than Manual operation at 32 genomic loci. Then, present a statistical method and calculation (KS-test?) to determine whether the difference between Manual Vs. high-throughput gene editing is statistically different.

-The tables in sections 'b' and 'e' could be moved to supplementary information as they are redundant to results presented in sections 'a' and 'd'.

-This figure benefits from having genuine sequence fragments and base editing examples to demonstrate the categories found (No Editing, <5%, 6-10%, 11-49%, >50%, etc.).

-We should be clear about the difference between the number of target sites changed as the base location within an endogenous genomic locus. Again, another cartoon figure could be added to this figure to represent the strategy behind the base edits aimed.

Response: Thank you for these constructive comments. It was our fault that the calculation of base editing efficiencies were not illustrated. We added the explanation in the revised manuscript.

“EditR software was used for quantifying the base editing efficiency from the Sanger sequencing data, which is a free online tool broadly employed in the field²¹. Using the Sanger sequencing files and corresponding gRNA sequences respectively, editing results can be acquired in batches by the processing of Python.” please see Page 6, Line 153-157.

The following equation was added to explain how base editing efficiencies were calculated during base editing.

$$\frac{\text{The fluorescence area of all T}}{\text{The total area of fluorescence of all the bases (A, T, C, G)}} = \text{Editing efficiency of CBEs \%}$$

As suggested, the 'b' and 'e' were moved to supplementary information, the details showed in Supplementary Table 1 and Table 2. we have added Statistical analysis between the base editing efficiency of Manual Vs. high-throughput gene editing in the revised manuscript, Statistical analyses were conducted using Graphpad Prism. P values less than 0.05 were considered statistically significant.

The previous Figure 3c (pie chart) has been modified to Figure 4b, which represents the overview for the base editing results of BE4max at 1210 endogenous target sites in HEK293T cell. As mentioned above, the 1210 pathogenic or likely pathogenic SNV sites were selected randomly, the popular BE4max editor was used to test the high-throughput editing platform. Only 20.74% target sites have not been edited or with extremely low efficiencies, 68.02% target sites were effectively edited. This result demonstrated that the automatic high-throughput platform for human cell was successfully established, thousands of endogenous target sites can be manipulated simultaneously and efficiently. According to previous literature of BEs, BE4max showed variable performance at different endogenous target sites. Thus, when the target sites changed, BE4max demonstrated different performance, and so as to the other base editors. We have added relevant content in the revised manuscript. "To demonstrate the capacity of the automated high-throughput system, a total of 1210 disease-associated SNVs were selected randomly as targets for editing with BE4max. Among the 1210 targets, 823 showed 10 to 50% C-to-T conversion efficiency, 248 targets had $\geq 50\%$ editing efficiency, and 136 targets showed 5 to 10% editing efficiency. We were unable to obtain editing results from 76 gene targets due to unsuccessful PCR amplification of the target loci, and 175 gene targets had less than 5% efficiencies (Fig. 4b). The editing efficiencies of the 1210 targets are listed in the Supplementary Data 9. The editing results of BE4max were in accordance with the previous reports, which showed greatly varied performance at different target sites²⁴." please see Page 7, Line 184-191.

(4) In Fig. 4:

-In Panel-ZERO (first one), can you have error bars (based on biological replicates) for the actual base editing efficiency for better comparison? Are the statistical methods able to generate the same for better comparison?

-In the same Panel-ZERO (the new one), I would add some examples of base editing using the BE4max and A3A-nCas9 methods to demonstrate their molecular mechanisms. A broader audience could benefit from having a better understanding of them.

-In panel-C, it is not clear what result or comparison should be made or expected. For example, some pair-wise comparisons could be between the actual and other model results indicated inside the plot to represent the method's performance developed in this work.

-In panel-B, please define N20s in the figure legend.

Response: Thank you for these constructive comments. The previous Fig. 4 has been modified to Fig. 5 in new manuscript.

Human APOBEC3A based BEs can mediate efficient C-to-T base editing in GpC dinucleotide regions. Thus, A3A-BE4max was used to edit the loci, which were converted with low efficiencies by BE4max. The results from A3A-BE4max indicated that it was the sequence preference of BE4max, but not the high-throughput method, responsible for the low efficiencies at some loci. Statistical analyses were conducted using the Graphpad Prism.

In panel-C, we selected ten endogenous targets sites from previous report, their editing outcome were predicted by our predict learning model (CAELM) and Be-Hive respectively. As shown in Fig. 5c, the results from CAELM with chromatin accessibility inputs was closer to the actual data. This figure was the comparison of the accuracy of two models, so there were no error bars for the predicted base editing efficiencies.

(5) In Fig. 5:

-Correct the position of the red rectangles where base locations are highlighted; some of them are misaligned and may create confusion.

-What is the takeaway message from this figure? Can you add the results you wish your audience to observe and compare, perhaps next to the red rectangles?

Response: Thank you for the constructive comments. To explain the editing figures more clearly, we have corrected and reworked the figure in the revised manuscript. In our revision, we transferred it to Supplementary Fig. 6, which shows the editing results of cell disease models during the generation and correction of pathogenic point mutations. A cartoon figure (Supplementary Fig. 6) was added to demonstrate the molecular mechanism for CBEs and ABEs. In Supplementary Fig. 6b, the caption below the picture “BE4max editing efficiency” means that BE4max yielded effective C-to-T base editing in HEK293T cell according to Sanger sequencing and the analysis of EditR. The “monoclonal of SNV cell model” means that edited cell pools were sorted to get homozygous mutants, the “ABE8e correction” means that correction of pathogenic point mutations were performed by ABEs, and Sanger sequencing was used to check the effect.

Supplementary Fig. 6 (a) The editing process of building disease models by BE4max and correction of SNV cell models by ABE8e.

4. In the Results section, please address the following points:

(1) The abstract presents the following sentence: 'And the chromatin accessibility parameter was calculated to take up 1/6 importance relative to the context-sequence.' I could not find any figure or table that could demonstrate this result itself or its relative importance to other parameters.

Response: Thank you for these constructive comments. We have added relevant content in the revised manuscript. The phrase was modified to the following: “Feature importance score is commonly used in assessing how much each input feature contributes to build a model and predict a target variable²⁹, which provides insight of determining which feature might be most relevant to a target. To determine the approximate contribution of the chromosomal environment condition to the prediction, relative to the contribution of the DNA sequence context, feature importance scores of the sequence context and the DNA accessibility value were obtained respectively from the different models. We used the built-in function `get_score()` of the XGBoost python package to get the feature importance scores. Instead of learning inputs including sequence information, melting temperature and GC contents¹³, we used 2 inputs which were the target sequence context and the DNA accessibility. By the means of one-hot encoding, the target sequence context was transformed into dummy features where each base generated 4 dummy features corresponding to the existence of A, C, G and T, with DNA accessibility as a feature by itself. Therefore, there were $4n+1$ features in total where n was the number of the bases, and the feature importance score was calculated for each of them. Finally, by dividing the feature importance of the DNA accessibility value by the sum of the importance value of the rest features of the sequence context, we obtained a ratio which was 1 :6.398, and the expanding models were also analysed, the calculated ratios between the contribution of the DNA accessibility and the DNA context were close to 1:6 respectively (Supplementary Table 4)”, please check in the result, Page 11, Line 303-322.

(2) First, in Results, the Pearson correlation is missing a citation. Second, the following phrase is not accurate and potentially statistically wrong: 'A reasonable correlation was found between the experimental and predicted values (Pearson's correlation coefficient $r = 0.64$), which suggests a decent prediction accuracy' What is the best way to describe these results in a statistical sense? What is a decent accuracy in a statistical sense or given some benchmark?
-How were 'feature importance scores' obtained from the model? Is this a method that has been published or a new method developed in this study?

Response: Thank you for the comment and question. Citation for Pearson's r was added in revision. The phrase was modified to the following: “CAELM displayed good ability to predict base editing efficiencies, with the Pearson's correlation values ranging from 0.42 to 0.86, Pearson's r is among the most prevalent metrics for evaluating the accuracy of models for numerical data³⁹ and used in many previous literatures^{13, 14} in studying base editing efficiencies. A r value of exactly 1 or -1 indicates that there is a perfect positive or negative linear dependency between the two features respectively, a r value greater than 0.5 normally implies a strong linear relationship. The machine learning models achieved r values ranging from 0.50 to 0.95 by Song et al.¹³, and around 0.7 to 0.9 by Arbab et al.¹⁴, which were higher than the r value of 0.64 we obtained with CAELM. However, it was reasonable since their models were trained with a greater number of deep sequencing reads compared to ours. But in contrast to previous machine learning models, the real chromosomal environment of the target sequence was considered in our model, which provided better and more relatively realistic prediction.”, please see Page 15-16, Line 444-456.

(3) Next, in the following result, 'To our best knowledge, this is the first quantitative assessment of the relative importance of the two major factors, DNA sequence and chromatin environment, for determination the genome editing efficiency of a genomic locus. This result suggested that

although the DNA sequence was the major influential factor, but the chromatin environment also determined the editing efficiency and could not be ignored.' If the method used to determine the influence of each factor in shaping the editing efficiency was created in this study, please add at least one more independent method (preferably known) to validate these results.

Response: Thanks very much for the great comment. We have added relevant content in the revised manuscript. The phrase was modified to the following: "Previous researches have shown a strong correlation between the performance of nuclease and the chromatin accessibility properties^{14, 16}. Kristopher et al. demonstrated that gene editing was more efficient in euchromatin than in heterochromatin¹⁶. Large-scale genetic screens in human cell lines indicated that highly active sgRNAs for Cas9 and dCas9 were found in regions of low nucleosome occupancy, and the nucleosomes directly impeded Cas9 binding and cleavage *in vitro*¹⁷. We previously found that pioneer factor, such as Vp64, improved CRISPR-based C-to-G and C-to-T base editing by changing local chromatin environment¹⁸. However, current studies on deep learning employed the editing data from lentiviral integrated target sequences, while the real chromosomal environment of the target sequence was ignored^{13, 14}", please see the introduction section.

Overall, the base conversion result is a combination of chromatin environment and sequence. "Based on the *in situ* genome editing data obtained by the automatic high-throughput platform, we developed a new Chromatin Accessibility Enabled Learning Model (CAELM), to predict the performance of cytosine base editors (CBEs). For the first time, the real chromosomal environment of the target sequences was considered in models building, which provided better prediction based on the realistic data.", please check in the abstract.

Since the predict-learning model we built in this work is the first one that utilized the two inputs of sequence and chromatin accessibility, we could not find one existing method to work like this one. And for the current stage, it is also beyond our capacity to come up with a parallel new model that could provide one more independent method to validate this model. We are very sorry. But we further enhanced the application value of CAELM, please check in the result, Page 10-11, Line 270-300. By training the core model with an additional relatively small set of *in situ* editing data with desired editors or cell lines, an expanded CAELM model with comparable or even better accuracy could be obtained, which indicated the CAELM prediction compatibility might be universally applicable with the expansion strategy.

5. For data availability, there could be repositories where a link for download could be provided. In any case, please make sure it follows the journal's guidelines.

Response: Thank you for these constructive comments. For data availability, The Sanger sequencing data from this study have been submitted to the NCBI Sequence Read Archive under accession number PRJNA885770. The data sets used in this study are provided as Supplementary Data 1-10 and Supplementary Table 1-5. The plasmids for expression of gRNAs and BEs are listed in Supplementary material (Supplementary Fig. 6-12). These Python scripts and the prediction models are available on GitHub (<https://github.com/YQLiCAS/BE4max>) for public researching usage.

Reviewer #2 (Remarks to the Author):

This paper studies in situ base editing efficiency, and provides a dataset of base editing efficiency measured at 1134 genomic loci in the native chromatin and epigenomic contexts. While the trained machine learning model is limited in scope, and its performance does not obsolete BE-Hive, I expect that the paper will be of interest to scientists interested in the effect of chromatin context on base editing efficiency.

1. The authors take a reasonable and standard approach in data processing and training the machine learning model. The paper's clarity could be improved by specifying the Sanger sequencing read count in the methods. The authors describe proper cross-validation for hyperparameter selection, and test set evaluation. However, the paper's clarity could be improved by reworking the caption for figure 4, and clarifying in the main text that the reported performance ($r=0.64$) is on the held-out 20%. The paper could be strengthened by performing full k-fold test set evaluation: if the test set is 20% of the full dataset, train and test 5 times on each non-overlapping split and report 5 test-set performances for the trained model and BE-Hive. This would enable readers to better evaluate the model's improvements over BE-Hive.

Response: Thanks very much for pointing out the mistake. The caption for figure 4 was changed to "Figure 5. The performance of the CAELM Models for prediction base editing outcome." . We used five-fold cross-validation to determine the optimal hyperparameters according to the suggestion. We have added relevant content in the revised manuscript: "The model predicting the base editing efficiencies of BE4max in HEK293T was trained and tested in an 80:20 ratio, meaning 80% targets information was used to train the model, and the remaining 20% was used for prediction testing. The accuracy of the model was measured by Pearson's correlation coefficient, and achieved a r value of 0.64 between the measured and predicted value (Fig. 5a). We also performed a 5x5 nested cross-validation²⁷, which is a gold standard for reliably performing hyperparameters tuning and performance assessment. We used an inner loop to select optimal hyperparameters by grid search and an outer loop to evaluate the performance of the model (Supplementary Fig. 5a). Concretely, the data was divided into 5 outer folds, each fold accounted for 20% and was left out as the testing fold. The remaining 80% samples were further split into 80% training and 20% validation set in the inner loop to obtain the optimized hyperparameters, which were used in the outer loop to retrain and evaluate the model, a Pearson's correlation coefficient, $r = 0.64$, was obtained as the average metric value across the five outer testing folds and suggested a good prediction accuracy (Supplementary Fig. 5b).", please see page8-9, line 232-247.

2. The machine learning model's contributions may be limited in scope, as it is trained on only one base editor (BE4Max) in one cell-type (HEK293T). As such, it is unlikely that the model will achieve broad use by the base editing community. This contribution could potentially be strengthened by proposing a machine learning approach to extrapolate from learned chromatin accessibility dependencies to other base editors or cell-types; such approaches could be evaluated using the small dataset of 30 genomic loci reported to be edited by both APOBEC3a-nCas9 and BE4Max in the paper.

Response: Thank you for these constructive comments. As suggested, to extend the prediction capability to the base editing result of other base editors and cell lines, we have added relevant

content in the revised manuscript: “To further enhance the application value of CAELM, in addition to BE4max, two other typical CBEs, Anc-BE4max and hyA3A-BE4max, were added in the model learning. To the effect of chromatin accessibility parameter in different cell types, the C-to-T conversion of about 100 endogenous target sites were analysed in HEK293T and HepG2 cells respectively (Supplementary Fig. 4). The experiment generated 5 corresponding data sets, including Anc-BE4max_HEK293T, hyA3A-BE4max_HEK293T, BE4max_HepG2, Anc-BE4max_HepG2 and hyA3A-BE4max_HepG2 (Supplementary Data 10).

To boost the fitted model on new data, we continued training via loading the predictive model of BE4max_293T using the parameter ‘xgb_model=’ in the model fitting process. Five new models including Anc-BE4max and hyA3A-BE4max in HEK293T, BE4max, Anc-BE4max and hyA3A-BE4max in HepG2 were built by continuing learning on the BE4max_293T model. The new data was continually trained on the BE4max_293T model and tested at an 85/15 ratio. Data of the target sequences shared with the test set was excluded from the data of BE4max in HEK293T to train the base model in the first place. Good correlations were achieved between the experimental and predicted efficiencies of Anc-BE4max (Pearson’s $r = 0.86$), hyA3A-BE4max in HEK293T (Pearson’s $r = 0.72$); and BE4max (Pearson’s $r = 0.70$), Anc-BE4max (Pearson’s $r = 0.87$), and hyA3A-BE4max (Pearson’s $r = 0.42$) in HepG2 respectively (Fig. 5d, Fig. 5e, Fig. 5f, Fig. 5g, Fig. 5h). Compared to the Pearson’s correlation of 0.64 of BE4max in HEK293T (Fig. 5a), the models of the new base editors and cell types retained good predictive ability with similar and even higher correlation values.

In the model expansion process, we constituted the strategy to adapt the core CAELM model to more types of CBE base editors in various cell types. By training the core model with an additional relatively small set of *in situ* editing data with desired editors or cell lines, an expanded CAELM model could be obtained. Theoretically the CAELM prediction compatibility was universally applicable with the expansion strategy.”, please see page10-11, line 272-300.

3. The paper's larger contribution may be its systematic study of the contribution of chromatin context to base editing efficiency. The analysis supporting the conclusion of 1/6 relative importance appears sound to me. However, the paper is lacking any mention or discussion of prior work that study the relevance of chromatin context to Cas9 editing. The paper would be strengthened if such comparisons or discussion were included.

Response: Thanks very much for the very constructive comment. We have added relevant content in the introduction of revised manuscript. We have added following content to clarify the current understanding of the relationship between BE outcomes and chromatin environment in introduction. “Previous researches have shown a strong correlation between the performance of nuclease and the chromatin accessibility properties^{14, 16}. Kristopher et al. demonstrated that gene editing was more efficient in euchromatin than in heterochromatin¹⁶. Large-scale genetic screens in human cell lines indicated that highly active sgRNAs for Cas9 and dCas9 were found in regions of low nucleosome occupancy, and the nucleosomes directly impeded Cas9 binding and cleavage *in vitro*¹⁷. We previously found that pioneer factor, such as Vp64, improved CRISPR-based C-to-G and C-to-T base editing by changing local chromatin environment¹⁸. However, current studies on deep learning employed the editing data from lentiviral integrated target sequences, while the real chromosomal environment of the target sequence was ignored^{13, 14}. One of the

reasons is that it is difficult to obtain a large set of editing data from endogenous target sites”.
please see page3, line 75-86.

Reviewer #3 (Remarks to the Author):

The authors present here an automated genetic editing platform combined with machine learning approaches to predict editing efficiency. The genetic editor involves a base editor, capable of changing a single base pair without the need to induce double-stranded DNA breaks. These types of genetic changes are of interest in mammalian cells because single nucleotide variations are known to be involved in a large fraction of human diseases. Hence, cell disease model cells are desirable for research purposes.

1. In my opinion, most of the value of this work lays in the production of the automated pipeline. Automating molecular biology is an arduous process and the claim of a fully automated process is very interesting. However, I have my qualms that the authors have truly achieved a “fully automated process” in the sense that no human intervention is needed from design to prediction, given the equipment description. It looks more like an automated pipeline in which humans transport cells and reagents from phase to phase. Merit worthy and interesting, but less impressive.
Response: Thank you for the helpful comments. Thank you for your suggestion. In this case, some steps of work still need human intervention. The “fully automated” of this manuscript was changed to “automated”. Meanwhile, in order to illustrate the automation process more clearly, we modified Fig. 1, and supplied the detail workflow of each module in supplementary Fig. 1. The inputs, outputs, and human intervention steps in modules are demonstrated in Supplementary Fig. 2.

2. The machine learning approach, using an existing single standard method (XGB) to leverage the large amounts of data generated (~ 1000 genetically edited cells) to predict editing efficiency is, in comparison, a rather less imposing effort. The results are better than a previously published neural net ML method (BE-Hive), but not by much (0.64 vs 0.52 Pearson correlation coefficient). The only real novelty is the use of the chromatin accessibility information as part of the input vector for the machine learning algorithm, despite claims to the contrary (“we developed a new machine learning model...”). The estimate of its relative importance in predicting the gene editing efficiency is novel and somewhat valuable (estimated to be ~1/6 of all importance scores).

In summary, I believe that the most valuable part of this work relies in the automated pipeline able to produce thousands of genetically edited model cells. I don't, however, work on mammalian cells or health, so I feel I am not in a good position to determine the overall value of the paper to the field. I will leave that to the experts.

Response: Thank you for these constructive comments. As suggested, we have added some detailed comparison in the article to demonstrate the advancement to a broader audience, who might not work in the field of mammalian cell genome editing. In summary, we have built an automated high-throughput genome editing platform for mammalian cells for the first time. To show the whole editing process in an unambiguous manner, as show in result and figure1, the automated platform was mainly constituted of four modules, including (1) computer-aided design of endogenous target gRNAs, (2) the construction of gRNA expression plasmids, (3) base editing in mammalian cell, and

(4) machine learning for CBEs performance model building, which is shown in Fig. 1. The detail of the content and figure was modified thoroughly.

To let other researchers to use our high-throughput methods, the data processing for high-throughput platform, the custom codes for the frontend and backend software used in this study are available on GitHub (<https://github.com/YQLiCAS/BE4max>) for public researching usage. The table list of data input and output are available in the Supplementary Information table1-7. Thus, the high-throughput genome editing platform has been demonstrated. Furthermore, the expense and time of high-throughput method Vs manual operations were compared, and the detailed comparisons were supplied in supplementary Fig. 3. High-throughput method is less time-consuming, and much less expensive when working on a large number samples.

To extend the prediction capability of our model to the base editing result of other base editors and cell lines, we have added relevant content in the revised manuscript: “To further enhance the application value of CAELM, in addition to BE4max, two other typical CBEs, AncBE4max and hyA3ABE4max, were added in the model learning. To the effect of chromatin accessibility parameter in different cell types, the C-to-T conversion of about 100 endogenous target sites were analysed in HEK293T and HepG2 cells respectively (Supplementary Fig. 4). The experiment generated 5 corresponding data sets, including Anc-BE4max_HEK293T, hyA3A-BE4max_HEK293T, BE4max_HepG2, Anc-BE4max_HepG2 and hyA3A-BE4max_HepG2 (Supplementary Data 10).

To boost the fitted model on new data, we continued training via loading the predictive model of BE4max_293T using the parameter ‘xgb_model=’ in the model fitting process. Five new models including Anc-BE4max in HEK293T, hyA3A-BE4max in HEK293T, BE4max in HepG2, Anc-BE4max and hyA3A-BE4max in HepG2 were built by continuing learning on the BE4max_293T model. The new data was continually trained on the BE4max_293T model and tested at an 85/15 ratio. Data of the target sequences shared with the test set was excluded from the data of BE4max in HEK293T to train the base model in the first place. Good correlations were achieved between the experimental and predicted efficiencies of Anc-BE4max (Pearson’s $r = 0.86$), hyA3A-BE4max in HEK293T (Pearson’s $r = 0.72$); and BE4max (Pearson’s $r = 0.70$), Anc-BE4max (Pearson’s $r = 0.87$), and hyA3A-BE4max (Pearson’s $r = 0.42$) in HepG2 respectively (Fig. 5d, Fig. 5e, Fig. 5f, Fig. 5g, Fig. 5h). Compared to the Pearson’s correlation of 0.64 of BE4max in HEK293T (Fig. 5a), the models of the new base editors and cell types retained good predictive ability with similar and even higher correlation values. The contribution of chromatin accessibility in prediction was determined referring to above method, taking up about 1/6 importance relative to the context-sequence (Supplementary Table. 4).

In the model expansion process, we constituted the strategy to adapt the core CAELM model to more types of CBE base editors in various cell types. By training the core model with an additional relatively small set of *in situ* editing data with desired editors or cell lines, an expanded CAELM model could be obtained. Theoretically the CAELM prediction compatibility was universally applicable with the expansion strategy”, please check in the result, page 10-11, line 272-300.”,

3.Finally, I would say that the paper is hard to read for non-specialists and needs some improvement in its grammar to be easily digested.

Response: Thank you for these constructive comments. As suggested, after new data was added and changes were made to the revised manuscript, we have carefully improved the English writing of the manuscript. We hope this revised version is good for publication.

Reviewers' Comments:

Reviewer #1:

Remarks to the Author:

I want to thank the authors for their thoughtful revision of the Manuscript. Namely, the authors have included new relevant method descriptions, changed Manuscript figures, added new Supplementary Figures, and performed further statistical analyses discussing them in supplementary results and tables accordingly. It looks and reads significantly clearer and better now. Most concerns and suggestions highlighted in my first revision have been scientifically and intelligently addressed. Therefore, I believe the Manuscript can be considered for publication in Nature Communications. I have no more concerns or suggestions at this point.

Editorial board questions

-What are the noteworthy results?

R. Yes.

-Will the work be of significance to the field and related fields? How does it compare to the established literature? If the work is not original, please provide relevant references.

R. Yes.

-Does the work support the conclusions and claims, or is additional evidence needed?

R. Yes.

-Are there any flaws in the data analysis, interpretation, and conclusions? Do these prohibit the publication or require revision?

R. Fairly. The work still lacks an independent validation for their central claim in the Manuscript. Still, since it is the first time anyone has created such a method, I do not think this is something that prohibits the Manuscript from being published.

-Is the methodology sound? Does the work meet the expected standards in your field?

R. Yes.

-Is there enough detail in the methods for the work to be reproduced?

R. Yes.

Reviewer #2:

Remarks to the Author:

The revised manuscript has addressed my concerns.

Reviewer #3:

Remarks to the Author:

The authors present here what seems to be the first automation approach to design, fabricate and test gene editing in mammalian cells in a high-throughput and automated way. Furthermore, this pipeline is used to assess the relative importance of chromatin accessibility information to predict editing efficiency, which is a useful insight.

The author have addressed my previous requests satisfactorily, except for one: the English used in the manuscript is still subpar. Specifically, the following corrections should be made before publishing:

Line 27: The acronym CAELM is used in the abstract without explanation, please add what it stands for and explain that it is an ML model for base editing performance prediction.

Line 29: "...of the target sequences was considered in models building..." should be "...of the target sequences was considered in the building of the model..."

Line 31: "...context-sequence decided the editing..." should be "...context-sequence influenced the

editing..."

Line 74: "However, currently such data was obtained from integrated editing loci which is highly limited" Please explain why it was highly limited.

Line 75: "Previous researches have shown..." should be "Previous research have shown..."

Line 407: "...high-throughput saved both time and cost, especially the time..." should be "...high-throughput saved both time and cost, particularly the former..."

Fig 1b has two typos: it should be "gRNA" instead of "gRAN" in the title, and it should be "extraction" instead of "extration" in the fourth pannel

Reviewer #3 (Remarks to the Author):

The authors present here what seems to be the first automation approach to design, fabricate and test gene editing in mammalian cells in a high-throughput and automated way. Furthermore, this pipeline is used to assess the relative importance of chromatin accessibility information to predict editing efficiency, which is a useful insight.

The author have addressed my previous requests satisfactorily, except for one: the English used in the manuscript is still subpar. Specifically, the following corrections should be made before publishing:

Line 27: The acronym CAELM is used in the abstract without explanation, please add what it stands for and explain that it is an ML model for base editing performance prediction.

Line 29: "...of the target sequences was considered in models building..." should be "...of the target sequences was considered in the building of the model..."

Line 31: "...context-sequence decided the editing..." should be "...context-sequence influenced the editing..."

Line 74: "However, currently such data was obtained from integrated editing loci which is highly limited" Please explain why it was highly limited.

Line 75: "Previous researches have shown..." should be "Previous research have shown..."

Line 407: "...high-throughput saved both time and cost, especially the time..." should be "...high-throughput saved both time and cost, particularly the former..."

Fig 1b has two typos: it should be "gRNA" instead of "gRAN" in the title, and it should be "extraction" instead of "extration" in the fourth pannel

Response: Thank you very much for pointing out our carelessness. We appreciate your constructive comments and suggestions and have revised our manuscript carefully and thoughtfully based on each point raised. The changes were highlighted with yellow color in the revised manuscript in responding to the reviewers' suggestions. Thanks again for the help.

The new manuscript's line 24: The acronym CAELM had add explanation, the phrase "we developed CAELM to predict the performance of cytosine base editors (CBEs)" was changed to "we developed a new Chromatin Accessibility Enabled Learning Model (CAELM) to predict the performance of cytosine base editors (CBEs)".

Line 29: "...of the target sequences was considered in models building..." was deleted in the new manuscript.

The new manuscript's line 26-27: "...context-sequence decided the editing..." was changed to "...context-sequence are utilized to build the model..."

The new manuscript Line 66-67: "However, currently such data was obtained from integrated editing loci which is highly limited" was changed to "currently such data was obtained from integrated editing loci which is lacked *in situ* information".

The new manuscript Line 68: "Previous researches have shown..." was changed to "Previous research has shown..."

The new manuscript's line 389: "...high-throughput saved both time and cost, especially the time..." was changed to "...high-throughput saved both time and cost, particularly the former..."

Fig. 1b has been modified, the title "gRNA..." instead of "gRAN..." in the title and "extraction" instead of "extration" in the fourth panel.